# 3D variability analysis reveals a hidden conformational change controlling ammonia transport in human asparagine synthetase

Adriana Coricello [1,9], Alanya J. Nardone[2], Antonio Lupia [3,10], Carmen Gratteri [1], Matthijn Vos[4], Vincent Chaptal [5], Stefano Alcaro [1,3] ✉, Wen Zhu [2] ✉, Yuichiro Takagi [6] ✉ & Nigel G. J. Richards [7,8] ✉

Advances in X-ray crystallography and cryogenic electron microscopy (cryo-EM) offer the promise of elucidating functionally relevant conformational changes that are not easily studied by other biophysical methods. Here we show that 3D variability analysis (3DVA) of the cryo-EM map for wild-type (WT) human asparagine synthetase (ASNS) identifies a functional role for the Arg-142 side chain and test this hypothesis experimentally by characterizing the R142I variant in which Arg-142 is replaced by isoleucine. Support for Arg-142 playing a role in the intramolecular translocation of ammonia between the active site of the enzyme is provided by the glutamine-dependent synthetase activity of the R142 variant relative to WT ASNS, and MD simulations provide a possible molecular mechanism for these findings. Combining 3DVA with MD simulations is a generally applicable approach to generate testable hypotheses of how conformational changes in buried side chains might regulate function in enzymes.

There is considerable general interest in understanding how enzyme motions might be linked to activity[1–5], although their precise contribution to catalysis remains the subject of debate[6,7]. Certainly, allosteric regulation of enzyme activity depends on correlated residue movements[8–10]. Computer-based simulations[11–14] also support the idea that altered dynamics can be correlated with improvements in the catalytic efficiency of designer enzymes as they undergo optimization by the methods of directed evolution[15,16]. In a similar fashion, hydrogen-deuterium exchange experiments show that local motions in a distal, solvent-exposed loop of soybean lipoxygenase are correlated with C-H activation energy barriers in the active site[17,18]. Protein motions appear to mediate rapid interconversion of thermally accessible conformational states in enzymes, although many of these states

are not catalytically competent[19]. Substrate binding can also bias the conformational ensemble of the free enzyme to favor structures in which reaction can take place[20,21], although this idea has mostly been developed from NMR studies on enzymes that possess only a single active site. Much less is known about how dynamical motions impact catalysis in enzymes that possess two, or more, active sites, such as glutamine-dependent amidotransferases[22–24] or tryptophan synthase[25]. Such multi-site enzymes are widely regarded as being too large for routine NMR-based studies of their motions in solution[26], although chemical shift perturbations have been used to elucidate the molecular basis of allosteric activation in heterodimeric enzyme complexes, such as imidazole-glycerol phosphate synthase[27]. Methyl-transverse relaxation-optimized spectroscopy (TROSY)[28], also has the potential to

[1]Dipartimento di Scienze della Salute, Università "Magna Græcia" di Catanzaro, Catanzaro, Italy. [2]Department of Chemistry & Biochemistry, Florida State University, Tallahassee, FL, USA. [3]Net4Science Academic Spin-Off, Università "Magna Græcia" di Catanzaro, Catanzaro, Italy. [4]NanoImaging Core Facility, Centre de Resources et Recherches Technologiques, Institut Pasteur, Paris, France. [5]Molecular Microbiology and Structural Biochemistry Laboratory, CNRS UMR 5086, University of Lyon, Lyon, France. [6]Department of Biochemistry & Molecular Biology, Indiana University School of Medicine, Indianapolis, IN, USA. [7]School of Chemistry, Cardiff University, Park Place, Cardiff, UK. [8]Foundation for Applied Molecular Evolution, Alachua, FL, USA. [9]Present address: Dipartimento di Scienze Biomolecolari, Università degli Studi di Urbino "Carlo Bo", Urbino, Italy. [10]Present address: Dipartimento di Scienze della vita e dell'ambiente, Università degli Studi di Cagliari, Cagliari, Italy. ✉e-mail: alcaro@unicz.it; wzhu@chem.fsu.edu; ytakagi@iu.edu; richardsn14@cardiff.ac.uk

yield dynamical information for large, multi-site enzymes if isotopically labeled methyl groups in valine, isoleucine or leucine residues can be introduced into the protein[29,30]. Given the technical difficulty associated with such experiments, however, computational methods have become increasingly important for gaining a molecular understanding of dynamics in multi-site enzymes[31,32]; recent examples include studies of imidazole-glycerol phosphate synthase[33] and tryptophan synthase[34,35].

Notwithstanding the power of modern simulation methods, the past decade has seen rapid advances in new experimental techniques for observing the structural dynamics of proteins[3,36–38]. More specifically, novel methods to analyze EM-derived maps[39] are providing new opportunities to obtain information on enzyme motions directly from the EM map[40–42]. Here, we use human ASNS, a Class II glutamine-dependent amidotransferase that mediates asparagine biosynthesis[43,44], as a model system to evaluate whether EM-derived maps can provide insights into conformational changes that might mediate function in multi-site enzymes.

Human ASNS is also of considerable biomedical interest, having been linked to metastatic progression in breast cancer[45], sarcoma cell proliferation[46], and decreased effectiveness of clinical treatments for acute lymphoblastic leukemia[47]. Moreover, ASNS variants are strongly correlated with impaired neural development in children[48], pointing to the importance of gaining an in-depth understanding of how single residue changes might affect activity by altering the conformational preferences and dynamics of the enzyme. Structural[49,50] and biochemical studies show that ASNS is built from two domains, each of which contains an active site that catalyzes one of the half-reactions needed for the overall conversion of aspartate to asparagine (Supplementary Fig. 1)[51–53]. As in all other glutamine-dependent enzymes, catalytic turnover requires that ammonia must move through an intramolecular tunnel connecting the active sites[24,54].

In this study we show how combining the analysis of EM-derived variable maps[55] with atomistic MD simulations[56,57] identifies the functional importance of a single residue, Arg-142, for ammonia translocation in human ASNS. This conclusion is bolstered by the EM structure and steady-state kinetic characterization of the R142I ASNS variant, in which Arg-142 is replaced by isoleucine (the cognate residue in the bacterial homolog AS-B). MD simulations provide qualitative insights into the molecular basis for these experimental observations and suggest how the presence of a key intermediate in the synthetase active site might impact tunnel continuity and hence catalytic turnover in the R142I variant. The molecular mechanisms for regulating ammonia transfer in ASNS appear to differ sharply from those operating in other Class II glutamine-dependent amidotransferases, in which intramolecular tunnels connecting the active sites are created by large conformational rearrangements that are induced by substrate binding[58,59].

## Results

### Structure determination of human ASNS by cryo-EM

Human ASNS was expressed in insect cells and purified following standard procedures[60,61] (see "Methods"). Single-particle images of unmodified, recombinant human ASNS (apo-ASNS) were collected and processed using cryoSPARC[62]. To test the native state of ASNS in an unbiased fashion, the blob particle picker option of cryoSPARC was used to select -1.2 million particles followed by several rounds of 2D classification as well as ab initio reconstructions, giving a final total of 12 initial reconstructions (Supplementary Fig. 2). Eleven of these reconstructions exhibited small densities (Supplementary Fig. 2d), which were further processed because they might correspond to a monomeric form of the enzyme (Supplementary Fig. 2f). We were not, however, able to converge these into a high-resolution map corresponding to the ASNS monomer. The twelfth reconstruction clearly resembled the head-to-head ASNS dimer seen in the X-ray crystal structure (Supplementary Fig. 3a)[41]. No initial 3D reconstruction

resembling the head-to-tail dimer (Supplementary Fig. 3b) seen in the X-ray structure of the bacterial homolog (AS-B)[50] was found. We also examined the use of template-based particle picking with the AS-B dimer as template, but this attempt merely picked up a large number of junk particles (Supplementary Fig. 4). It is, therefore, unlikely that particles resembling the head-to-tail dimer exist; indeed, 3D reconstructions of particles selected by template-based particle picking give what appears to be a monomeric form of human ASNS when the AS-B dimer is the template (Supplementary Fig. 4). Taken together, these findings are consistent with human ASNS forming a head-to-head rather than a head-to-tail dimer. Realizing that the EM map of ASNS was similar to the crystal structure (Supplementary Figs. 2d and 3a), template-based particle picking was used to pick good particles efficiently. Data processing (see "Methods") eventually gave rise to a map with 3.5 Å overall resolution (Supplementary Table 1 and Supplementary Figs. 2, 5, 6 and 7). The model derived from the EM map (Fig. 1) was generated by utilizing our previous crystal structure of DON-modified WT human ASNS (6GQ3)[49].

Our EM structure answers several questions concerning the validity of the high-resolution X-ray crystal structure for human ASNS[49] in which the Cys-1 side chain is modified by 6-diazo-5-oxo-L-norleucine (DON)[63]. First, two monomers of unmodified human ASNS in the native state form a head-to-head dimer as a result of intermolecular interactions involving residues 31–34 in the adjacent N-terminal domains. The antiparallel orientation of these two peptide segments places the side chains of Arg-32 and Glu-34 in monomer A so that they can form salt bridges with Glu-34 and Arg-32 in monomer B, respectively (Fig. 1d and Supplementary Fig. 8). Moreover, Phe-31 and Phe-33 in both monomers form a hydrophobic cluster on the other face of the dimerization motif that likely contributes to the stability of the dimer interface in water. These contacts between the N-terminal domains of the ASNS monomers are almost identical to those observed in the X-ray structure, except the intermolecular disulfide bond is absent[49]. This important finding supports that the head-to-head dimer is likely present in the reducing environment of the cell. We conclude that head-to-head dimerization is an intrinsic property of the human enzyme, in contrast to the head-to-tail dimerization observed for the bacterial homolog. Second, the EM and X-ray structures of human ASNS both lack density for residues in two loop segments (residues 201-220 and residues 465-475) and the C-terminal tail (residues 539-560 and 536-560 in the EM and X-ray structures, respectively) (Supplementary Fig. 9). SDS-PAGE characterization of the human ASNS samples used in these cryo-EM studies shows no evidence for the presence of enzyme fragments (Supplementary Fig. 10). Given that the apo-ASNS used to obtain the EM structure was freshly prepared and vitrified immediately to make EM grids, there was insufficient time for proteolysis to take place. Thus, it is likely that these regions are disordered rather than being absent due to proteolysis as originally proposed[49].

Finally, amino acid side chains pointing into the tunnel (Val-119, Val-141, Arg-142, Leu-255, Met-344, Ala-404 and Leu-415) are clearly evident in the EM structure (Fig. 2a). These side chains are predominantly hydrophobic, consistent with the idea that ammonia is translocated between the active sites in its neutral form. The presence of a polar side chain (Arg-142) at the end of the tunnel close to glutaminase site is therefore unexpected. Arg-142 is, however, conserved in mammalian asparagine synthetases (Supplementary Fig. 11). Ser-362 and Glu-364, which are both located in the part of the tunnel adjacent to the synthetase active site, are also conserved residues (Supplementary Fig. 11). As observed for the X-ray structure of DON-modified human ASNS, the intramolecular tunnel in the EM structure adopts a discontinuous, form through which ammonia cannot diffuse (Fig. 2c and Supplementary Fig. 12). In contrast, the intramolecular tunnel is continuous when glutamine and AMP are bound within the active sites of the C1A AS-B variant (Supplementary Fig. 13)[50]. Given that the covalent modification of Cys-1 does not lead to stabilization of the

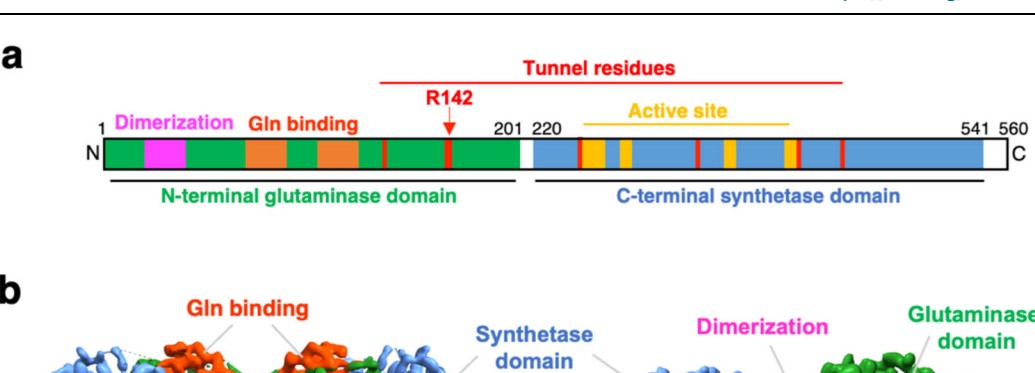

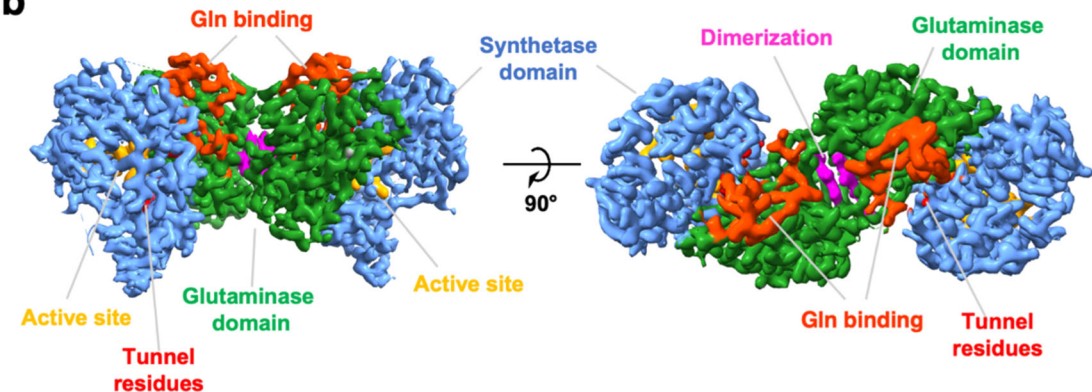

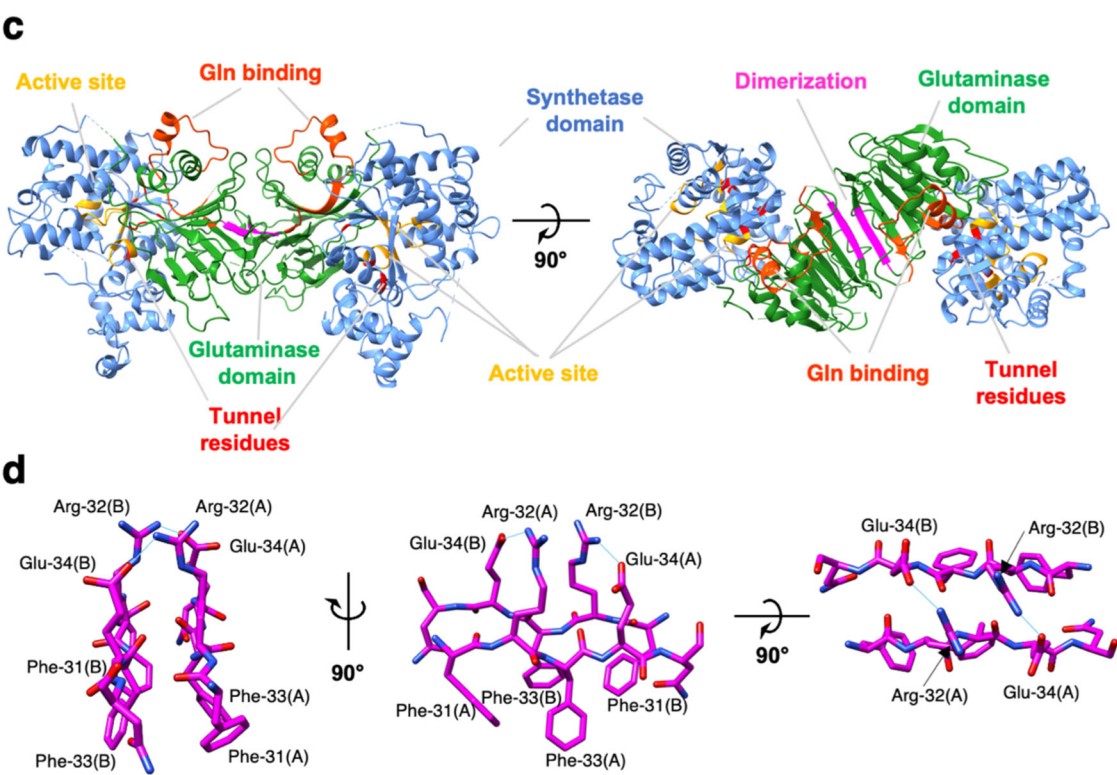

**Fig. 1 | Cryo-EM structure of human ASNS. a** Schematic representation showing the N-terminal glutaminase (forest green), dimerization (magenta) and the C-terminal synthetase (cornflower blue) domains. The glutamine-binding (orange red) and synthetase active site (orange) are also indicated together with residues constituting the intramolecular tunnel (red). **b** EM map and (**c**) ribbon representation of human ASNS (coloring is identical to that used in (**a**)). **d** Close-up view of the dimerization motif showing residues (C, magenta; N, blue, O, red) participating in salt bridges (shown by cyan lines), and hydrophobic interactions. (A): monomer A; (B): monomer B.

continuous form of the tunnel, our EM-based observations agree with previous proposals that AMP bound in the synthetase active site is important for facilitating ammonia translocation[50,64].

**Extracting conformational information for WT human ASNS from the cryo-EM map**
Competition experiments show that ammonia released from L-glutamine is not released into solution during L-asparagine synthesis

and must therefore travel along the intramolecular tunnel[54]. This translocation cannot take place when the tunnel is discontinuous, as seen in the EM and X-ray structures of recombinant human ASNS, suggesting that tunnel structure and function might be regulated by conformational effects and, perhaps, ligand binding or the formation of intermediates in the catalytic cycle. To investigate whether this is the case for the unliganded WT enzyme, we applied 3D variability analysis (3DVA)[65,66] to the EM map of apo-ASNS. As detailed

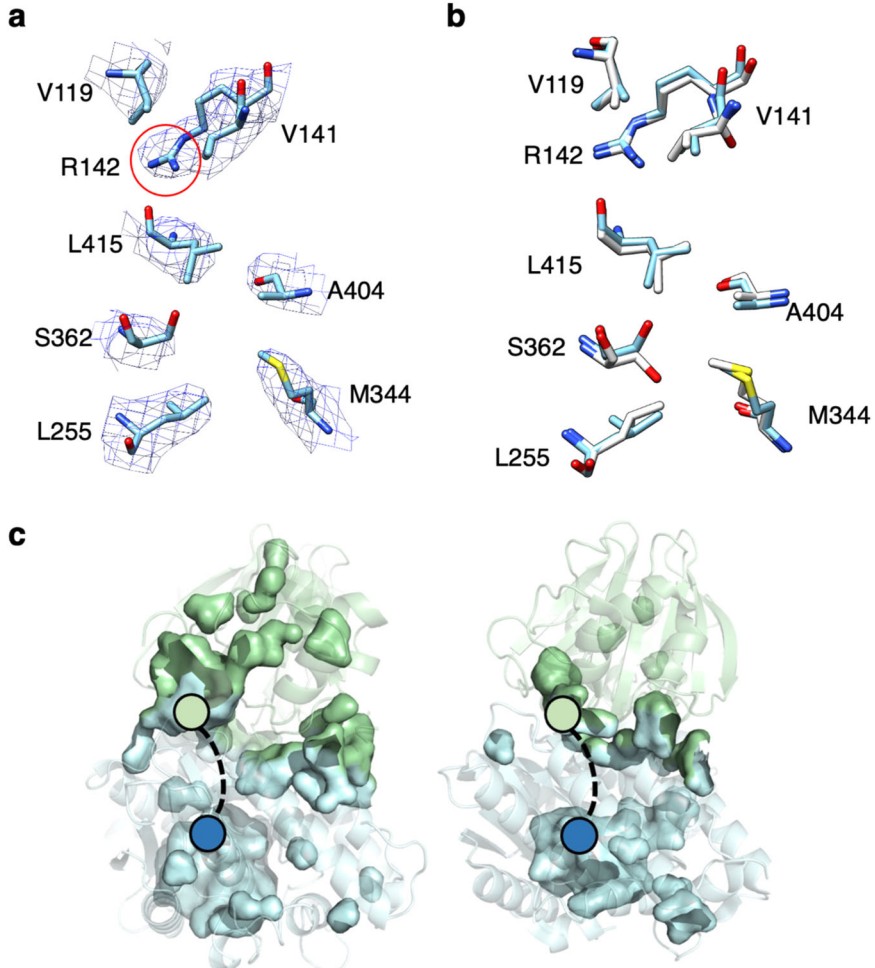

**Fig. 2 | Residues defining the intramolecular tunnel in WT ASNS as observed by EM and X-ray crystallography. a** Residues defining the intramolecular tunnel in the EM structure of WT ASNS with corresponding densities. Residues V119, V141, R142, L255, M344, S362, A404, and L415 are colored in sky blue and EM density (mesh) in blue. **b** Superimposition of residues defining the intramolecular tunnel in the EM (colored sky blue as in Fig. 1a) and X-ray (white) structures of human ASNS.

**c** Internal cavity and pocket representations, as computed by PyMol, showing the discontinuous tunnels in the EM (left) and X-ray (right) structures. The N- and C-terminal domains of both structures are colored green and blue, respectively, with colored circles showing the two active sites. Dashed lines connecting the active sites represent the position of the open tunnel seen in the X-ray crystal structure of the bacterial enzyme (Supplementary Fig. 13).

elsewhere[65], local differences in the protein structure are preserved by fast freezing, thereby leading to particles in which the enzyme may be captured in slightly different conformational states if any exist. As a result, 3DVA can identify conformational changes from the variance of particle stacks relative to a volume along a given principal component (PC) axis (see Supplementary Figs. 12 and 14). Although many residues throughout the protein might adopt alternate conformations in each PC, we were especially interested in those affecting the intramolecular tunnel involved in catalytic function. Analyzing the map of apo-ASNS along five PCs allowed us to convert the ensemble of EM densities into 100 high-resolution model structures (20 for each component), which revealed conformational changes in the Arg-142 side chain (Fig. 3 and Supplementary Fig. 15). We were, therefore, able to observe side chain movements in apo-ASNS at high-resolution despite these being much smaller than domain motions seen previously using 3DVA to study multidrug ABC exporters[67]. In our overall EM model (Fig. 3a), Arg-142 adopts a conformation in which the side chain blocks ammonia access to the tunnel, which we therefore designated as closed (Supplementary Fig. 15). This conformation is also seen in two of the 3DVA-derived PCs (components 0 and 2) (Fig. 3b, d). In two other PCs (components 1 and 3), however, the Arg-142 side chain is re-positioned to yield an alternate conformation, designated as open, which would allow

ammonia access to the tunnel (Fig. 3c, e). In the fifth PC (component 4), the side chain adopts a third conformation, designated as partially open (Fig. 3f). Closer examination of these structures showed that the carboxylate side chain of Asp-405 formed a salt bridge with Arg-142 when the latter adopted the closed conformation (Supplementary Fig. 15). The open orientation of Arg-142 is stabilized by a network of intramolecular interactions, involving Glu-76, Arg-142, Glu-414 and Asn-74 (Supplementary Fig. 15). Based on these findings, we hypothesized that Arg-142 might function as a gate to the ammonia tunnel thereby regulating ammonia translocation between the glutaminase and synthetase active sites. We also speculated that Arg-142 might play a role in the molecular mechanisms that convert the tunnel from a discontinuous form, seen in the X-ray structure of DON-modified human ASNS[49], to the continuous form that is present in the X-ray structure of the C1A AS-B variant complexed to glutamine and AMP[50]. So far, we have been unable to obtain cryo-EM structures of human ASNS bound to ligands or catalytic intermediates, such as the ASNS/β-aspartyl-AMP/MgPP$_i$ ternary complex, with which to test these hypotheses. In addition, the absence of several loop segments in our cryo-EM structure of the enzyme precludes the use of software tools for mapping the location and continuity of the intramolecular tunnel in the experimental structure.

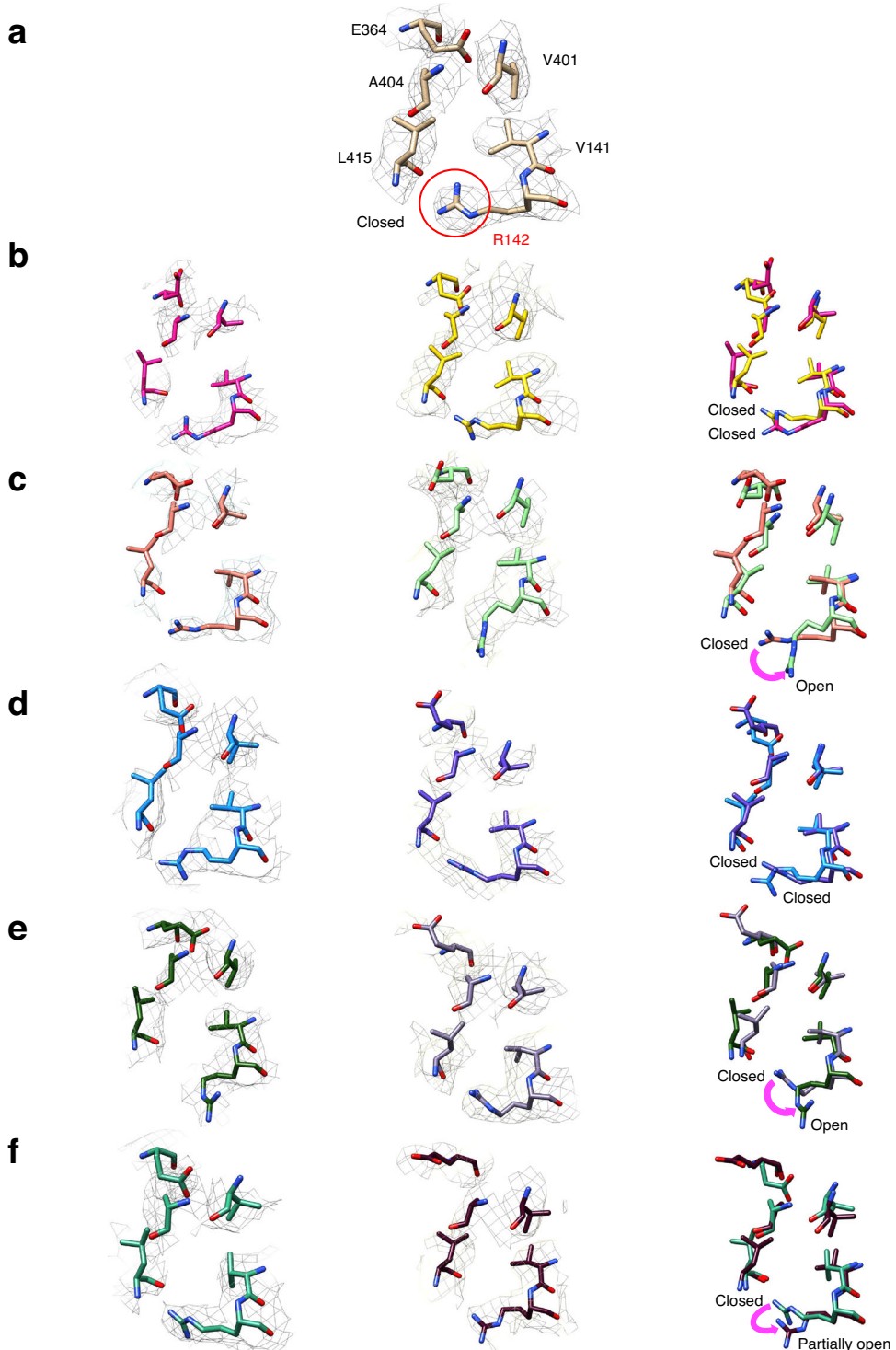

**Fig. 3 | PCA-derived structures of the ammonia tunnel in human ASNS obtained from 3D variability analysis. a** Side view of critical residues (Val-141, Arg-142, Glu-364, Val-401, Ala-404, Val-414, Leu-415) constituting the ammonia tunnel derived from the consensus EM map. The side chain of Arg-142 that blocks the tunnel is indicated (red circle). Side view of critical residues constituting ammonia tunnel derived from the variable EM maps derived from 3DVA and corresponding models generated by 3D variable refinement. Critical residues constituting ammonia tunnel derived from the maps at frame 1 (one end) and frame 20 (the other end) are superimposed: **b** PCA component 0, **c** PCA component 1, **d** PCA component 2, **e** PCA component 3, **f** PCA component 4.

## MD simulations of the full-length apo-ASNS monomer

To overcome these problems with testing the proposed functional role of Arg-142 and surrounding residues (Asp-405, Glu-76, Arg-142, Glu-414 and Asn-74)[68], we turned to computer-based approaches[40,69]. Prior work by Schulten[70–72] and Tama[73,74] shows the utility of MD simulations in interpreting cryo-EM maps. Computational strategies have also been

used to study conformational changes and the energetics of ammonia translocation in other glutamine-dependent amidotransferases[75–80], with the exception of any ASNS homolog.

An initial model of the full-length, human apo-ASNS monomer was constructed using the ROBETTA server (Supplementary Fig. 16a)[81]. Three segments in this model, corresponding to two loops (residues

210-221 and 466-478) and the C-terminal tail (residues 535-560), were computationally predicted because they are not observed by either cryo-EM or X-ray crystallography[49]. Published work suggests that ROBETTA provides accurate conformational predictions for missing loops[82], although the placement of the flexible C-terminal segment is less certain (Supplementary Fig. 17). Nevertheless, this computational model was not biased by data from the experimental cryo-EM studies.

The ROBETTA-derived apo-ASNS model was equilibrated in a box of explicit water molecules and the resulting system used in four replicate MD simulations of 200 ns duration at 298.15 K (Supplementary Fig. 18). First, we assessed whether these MD simulations of the apo-ASNS monomer were consistent with the 3DVA-derived EM variable structures. The root mean square fluctuations (RMSF) of backbone heavy atoms in all 100 of the 3DVA-derived EM structures were calculated using CPPTRAJ[83] and compared with those computed from structures sampled in the replicated MD trajectories of the WT apo-ASNS monomer. RMSF values were normalized to the largest value in each dataset before being used in this comparison, with the goal of offsetting differences in the magnitude of movements seen in the MD simulations and those based on the restrained coordinates used to construct the EM map (Fig. 4a). This comparison confirms that residue fluctuations seen in the MD trajectories generally correspond to those revealed by 3DVA, with the exception of residues adjacent to missing segments in the EM structure for which there are significant differences between experiment and simulation. For example, 3DVA-derived fluctuations seen for residues 207-209, 221-223, 464 and 478-479 (Fig. 4a) are associated with the poor quality of the EM map in these regions; the disagreement between theory and experiments in these regions is therefore artefactual. In a similar manner, the large fluctuations of the C-terminal tail in the replicated MD trajectories likely result from these residues being poorly positioned in the initial ($t = 0$ ns) model (Supplementary Fig. 17).

Second, we assessed the energetics of interconverting the conformers of Arg-142 within the apo-ASNS monomer. A free energy surface describing rotations of the Arg-142 side chain conformations was

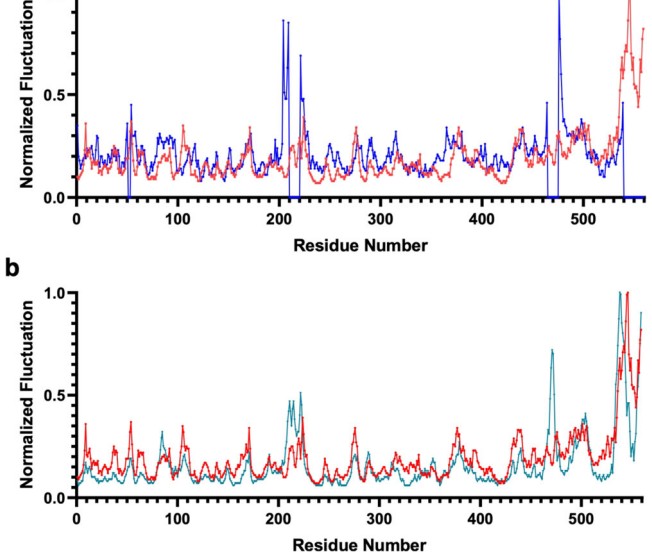

**Fig. 4 | Normalized RMSF values for residues in human ASNS. a** Comparison of values from the 100 3DVA-derived structures of chain A (blue) and the MD trajectory of apo-ASNS (red). As discussed in the text, the large 3DVA-derived fluctuations for residues 207-209, 221-223, 464 and 478-479 can be ignored in the comparison. **b** Comparison of values from the MD trajectories of apo-ASNS (red) and the human ASNS/β-aspartyl-AMP/MgPP$_i$ ternary complex (teal). Source data are provided as a Source Data file.

constructed using well-tempered metadynamics (WT-MTD) simulations[84,85]. Such calculations are well established in computational biophysics[86] and are suited to understanding the conformational preferences of proteins and peptides[87,88]. In these WT-MTD calculations, we chose the distance between the centers of mass of the guanidinium and carboxylate functional groups in Arg-142 and Asp-405, respectively, as one of the collective variables (CV1). A second collective variable (CV2) was defined to be the distance between the centers of mass of the guanidinium and carboxylate functional groups in Arg-142 and Glu-414. The calculated free energy surface shows a large free energy minimum (0.0 kcal/mol) suggesting that the Arg-142 side chain can adopt a range of conformations in the apo-enzyme (Supplementary Fig. 19a). In addition, a minimum corresponding to the closed conformation (CV1 = 5.4 Å and CV2 = 3.9 Å) is also present, which is only 1.2 kcal/mol higher in energy (Supplementary Fig. 19a). The conclusion that the conformational preference of Arg-142 is not biased by sampling or the force field is bolstered by distance measurements showing that the Arg-142 side chain undergoes multiple conformational interconversions over the course of a microsecond MD simulation (Supplementary Fig. 20a and Supplementary Table 2).

Finally, we assessed the relationship between conformational interconversions of Arg-142 and the structure of the intramolecular tunnel using four 200 ns replicate MD simulations of the apo-ASNS monomer model in water (Supplementary Fig. 18). The location and continuity of the intramolecular tunnel in four snapshots taken from one of these MD trajectories of the computational model were then analyzed using the CAVER 3.0 software[89]. In the initial snapshot ($t = 20$ ns), the tunnel is discontinuous and persists in this form for over 80 ns of simulation in apo-ASNS (Fig. 5a) but becomes continuous by 140 ns into the trajectory. These findings are generally supported by the replicate simulations.

With these snapshots in hand, we calculated the distance between the Arg-142 side chain and adjacent residues seen in the cryo-EM structure (Asn-74, Glu-76, Asp-405 and Glu-414) (Supplementary Fig. 21) to see if tunnel continuity and movement of the Arg-142 side chain were correlated (Supplementary Table 3). The measured distances in the four snapshots indicate that this is not the case, at least on the basis of our MD simulations. We therefore conclude that the gating function of Arg-142 is separate from the molecular events that form the continuous form of the tunnel; ammonia travels to the synthetase site only if the tunnel is continuous and the Arg-142 side chain adopts the open conformation.

## MD simulations of the human ASNS/β-aspartyl-AMP/MgPP$_i$ ternary complex

Recognizing that ligand binding might impact residue motions seen in the apo-ASNS monomer, we addressed these same questions using MD simulations of the ASNS/β-aspartyl-AMP/MgPP$_i$ ternary complex under identical conditions. Ammonia transport should be facilitated in this intermediate complex if β-aspartyl-AMP is to be converted into asparagine; it has been shown for AS-B, however, that β-aspartyl-AMP can be formed in the absence of glutamine[64]. These calculations employed a computational model that was built by docking the ligands β-aspartyl-AMP and MgPP$_i$ into the structure of WT apo-ASNS used in the MD simulations discussed above (see Methods and Supplementary Fig. 16b). As for the human apo-ASNS monomer, we performed well-tempered metadynamics (WT-MTD) simulations[85,86] to obtain a free energy surface describing the motions of the Arg-142 side chain when β-aspartyl-AMP is present in the synthetase active site. The relatively featureless surface seen for apo-ASNS becomes more complex (Supplementary Fig. 19b) with new energy minima corresponding to the closed, partially closed, and open conformations (Fig. 3 and Supplementary Fig. 15) being clearly defined. In the ternary complex, however, the open conformation is slightly preferred in energy (0.2 kcal/mol) to the closed conformation, and the barrier directly separating these

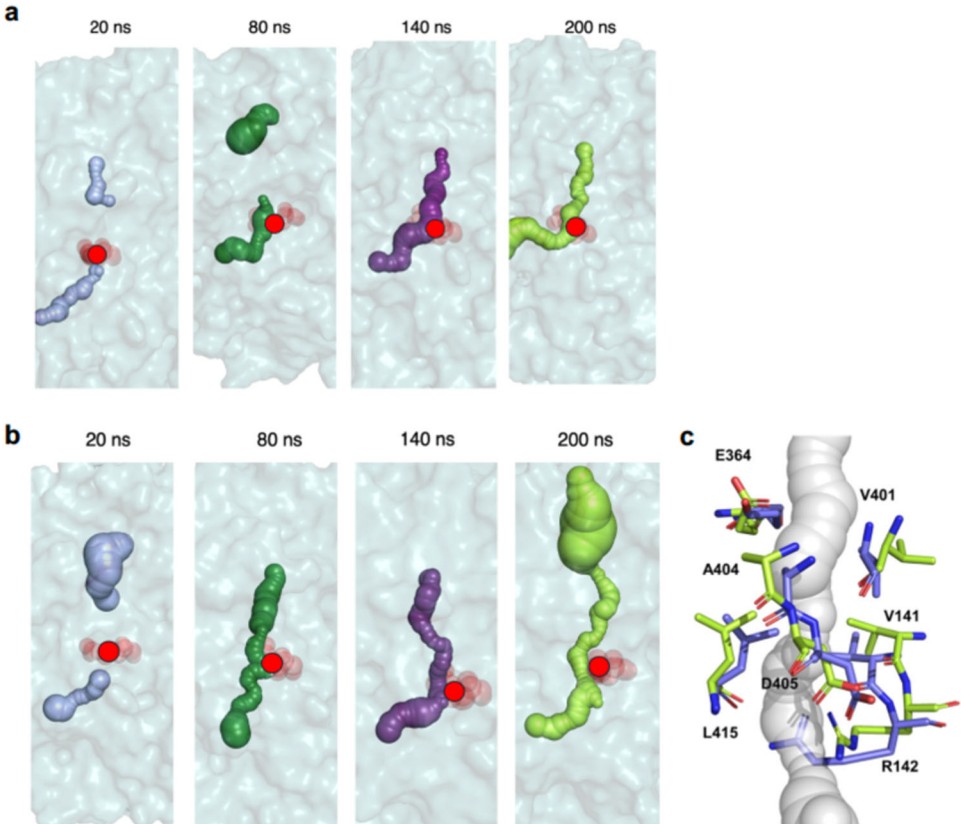

**Fig. 5 | Continuity of the intramolecular tunnel of WT human ASNS as observed by MD simulation.** Snapshots from the MD trajectories computed for (**a**) the WT apo-ASNS and (**b**) the WT ASNS/β-aspartyl-AMP/MgPPi ternary complex. The surface of the tunnel is indicated by spheres ($t = 20$ ns, light purple; $t = 80$ ns, forest green; $t = 140$ ns, violet purple; $t = 200$ ns, lime). Red circles indicate the position of the side chain of Arg-142 in the structures. Protein atoms are rendered as surfaces in cyan. Tunnels were identified using the CAVER 3.0 software package[89] and

visualized in PyMol. **c** Superimposed snapshots from the MD trajectory of the WT ASNS/β-aspartyl-AMP/MgPPi ternary complex showing the reorientation of the Arg-142 side chain (carbons are colored in slate ($t = 0$ ns) or green ($t = 200$ ns), and oxygen and nitrogen atoms are colored red and blue, respectively). Distances between the Arg-142 side chain and adjacent residues in these trajectories are provided elsewhere (Supplementary Tables S2 and S3).

two minima is calculated to be 2.1 kcal/mol. This result suggests that interconversion of the Arg-142 side chain conformation in the ternary complex is possible at 298 K. A third minimum is also seen, which is 1.0 kcal/mol higher in energy than that calculated for the open conformation (Supplementary Fig. 19b).

As in simulations of the apo-ASNS monomer, Arg-142 undergoes multiple interconversions between the open and closed conformations in a microsecond MD trajectory of the human ASNS/β-aspartyl-AMP/MgPPi ternary complex (Supplementary Fig. 20b), supporting the idea that Arg-142 can function to gate ammonia access to the tunnel. Under identical simulation conditions to those used for the apo-ASNS monomer, however, the tunnel in the human ASNS/β-aspartyl-AMP/MgPPi ternary complex adopts a continuous, open form within 80 ns (Fig. 5b and Supplementary Fig. 18b); findings that are consistent with replicate simulations (Supplementary Fig. 22 and Supplementary Table 4). Similar to apo-ASNS, measurements of the distances between the Arg-142 guanidium moiety and the side chains of Asn-74, Glu-76, Asp-405 and Glu-414 in the four snapshots are not correlated with formation of the continuous tunnel (Supplementary Table 5). On the other hand, a detailed analysis of the trajectory for the human ASNS/β-aspartyl-AMP/MgPPi ternary complex does indicate that the presence of the catalytic intermediate in the synthetase site disrupts a network of salt bridges involving Arg-142 due to the carboxylate of β-aspartyl-AMP forming a new interaction with the Arg-403 side chain (Supplementary Fig. 23). As a result, Asp-405 and Glu-414 can both adopt different conformations to those seen by cryo-EM, which stabilize the

open orientation of the Arg-142 side chain (Fig. 5c, Supplementary Fig. 22). In other words, the presence of the catalytic intermediate gives rise to a new network of salt bridges that is different to that observed in the EM-based structure of apo-ASNS (Supplementary Fig. 15).

## MD simulations of the full-length apo-R142I ASNS variant monomer

Arg-142 is conserved in mammalian asparagine synthetases, Sequence and structural alignments, however, show that that cognate residue in *Escherichia coli* AS-B is isoleucine (Ile-143) even though nearby residues are conserved in both homologs (Supplementary Fig. 11). We therefore used MD simulations to investigate the effects of replacing the polar arginine by a non-polar isoleucine on tunnel structure. The initial model of the full-length monomer of the apo-R142I ASNS variant was built from the energy minimized ROBETTA-based model of apo-ASNS described above. Following identical procedures to those described above, we observed that the tunnel remained continuous over the course of the 200 ns MD simulation with full ammonia access (Fig. 6a). This contrasts with our observations for WT apo-ASNS (Fig. 5a) and may merely be a consequence of the reduced length of the isoleucyl side chain. Distance measurements in the four replicate MD trajectories, however, indicate that Ile-142 side chain undergoes far less movement than that of Arg-142, perhaps to avoid intermolecular interactions with the polar carboxylate groups of Asp-405 and Glu-414 (Supplementary Fig. 21). Comparing the RMSF values for residues in

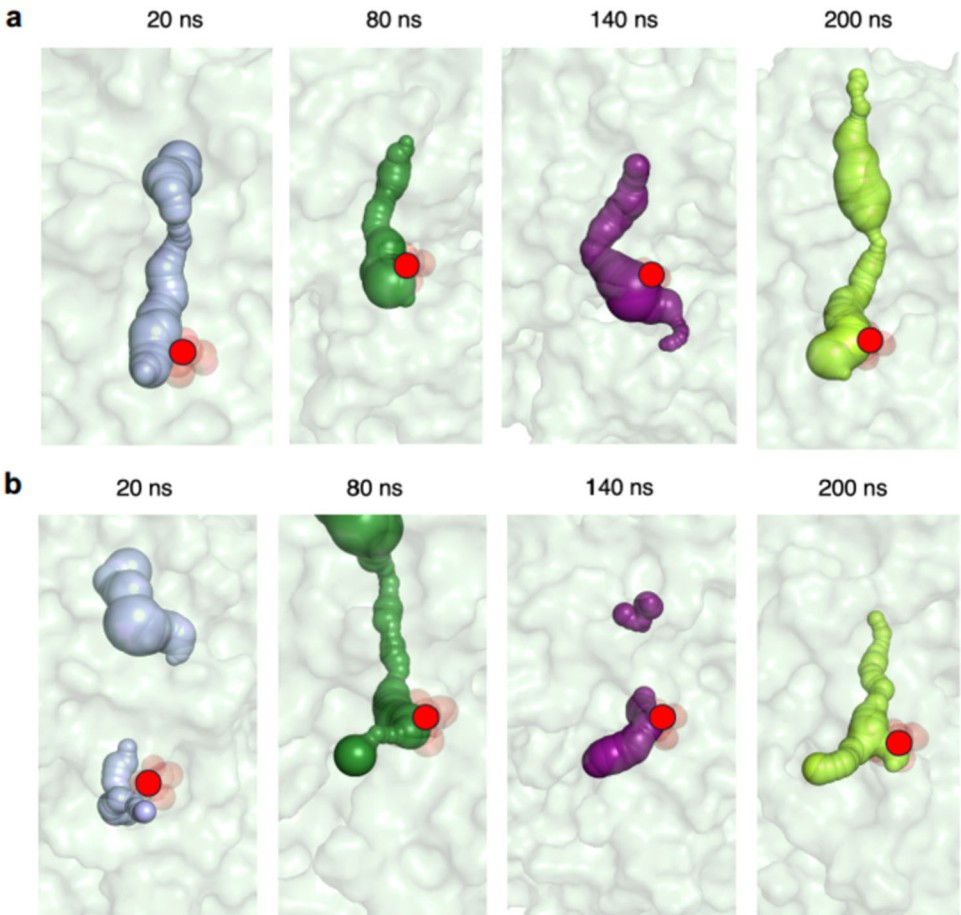

**Fig. 6 | Continuity of the intramolecular tunnel of the R142I ASNS variant as observed by MD simulation.** Snapshots from the MD trajectories computed for (**a**) the R142I variant and (**b**) the R142I/β-aspartyl-AMP/MgPPi ternary complex. The surface of the tunnel is indicated by spheres ($t$ = 20 ns, light purple; $t$ = 80 ns, forest green; $t$ = 140 ns, violet purple; $t$ = 200 ns, lime). Red circles indicate the position of the side chain of Ile-142 in the structure. Protein atoms are rendered as surfaces in cyan. Tunnels were identified using the CAVER 3.0 software package[89] and visualized in PyMol.

the R142I variant and the WT enzyme shows that the isoleucine side chain has a relatively small impact on the overall dynamics of the apo-enzyme (Supplementary Fig. 24a).

### Structure determination of the human R142I ASNS variant by cryo-EM

We were able to determine the structure of the R142I variant dimer at 3.35 Å resolution by cryo-EM (Supplementary Table 1 and Supplementary Figs. 25–28) in order to assess the validity of these MD simulations. As expected, replacing a polar group by a hydrophobic side chain had little impact on the overall structure of the enzyme (0.483 Å RMSD of WT vs. the R142I variant) (Supplementary Fig. 26) although structural changes in the tunnel region are observed that will be the subject of future computational analysis. The N-terminal entrance to the tunnel linking the two active sites is open in the EM structure, similar to that seen in the X-ray structure of the C1A AS-B variant bound to glutamine and AMP (Supplementary Fig. 29). 3DVA studies of the EM map of the R142I variant showed that the Ile-142 side chain adopts only a single conformation (Supplementary Fig. 30). Remarkably, this single amino acid substitution appears to rigidify the entire protein, as indicated by a 2-fold reduction for the observed particle distribution determined by 3DVA (compare Supplementary Figs. 14, 31).

### Steady-state kinetic measurements

Both the EM structure and MD simulations suggest that replacing Arg-142 by isoleucine results in unimpeded access of ammonia to a structurally continuous tunnel, coupled with rigidification of the R142I variant in the absence of ligands. Steady-state kinetic assays were therefore performed to determine the functional effects of replacing Arg-142 by isoleucine. In these experiments, the MgATP concentration was fixed, and activity was measured by detecting the formation of inorganic pyrophosphate ($PP_i$) production[90], which, by analogy to the bacterial homolog, is formed in a 1:1 stoichiometry with L-asparagine by WT ASNS[64]. We also investigated the activity of the R142A ASNS variant, in which Arg-142 is replaced by alanine. This variant was chosen because the alanine side chain is small. As a result, the gate of the tunnel should be open in this variant because Ala-142 is unable to form intramolecular interactions with other residues in the enzyme. When ammonia was used as the nitrogen source, varying the ammonia concentration gave steady-state parameters for the two variants that were essentially unchanged compared to those of WT ASNS (Table 1 and Supplementary Fig. 32). When L-aspartate was varied at saturating levels of L-glutamine, but in the absence of ammonia, the kinetics of PPi formation showed 4- and 2-fold reductions in turnover number for the R142I and R142A variants, respectively. In contrast, varying L-glutamine concentrations at saturating concentrations of L-aspartate gave 8- and 5-fold reductions in turnover number for the R142I and R142A ASNS variants, respectively, even though a substantial decrease in $^{app}K_M$ for L-glutamine is observed.

As the length of the intramolecular tunnel in human ASNS is ~20 Å and translocation is effectively a one-dimensional diffusion process[23], it is unlikely that moving ammonia between the two active sites is rate-

**Table 1 | Steady-state kinetic parameters for WT human ASNS and two variants**

| Nitrogen source | L-Glutamine | | | | Ammonia | |
|---|---|---|---|---|---|---|
| Enzyme | $^{app}K_M$ (Asp) | $k_{cat}$ | $^{app}K_M$ (Gln) | $k_{cat}$ | $^{app}K_M$ (NH$_3$) | $k_{cat}$ |
| | mM | s$^{-1}$ | mM | s$^{-1}$ | mM | s$^{-1}$ |
| WT | 0.25 ± 0.06 | 0.83 ± 0.04 | 0.9 ± 0.1 | 0.49 ± 0.02 | 0.5 ± 0.2 | 0.59 ± 0.05 |
| R142I | 0.09 ± 0.03 | 0.18 ± 0.01 | 0.01 ± 0.01 | 0.06 ± 0.01 | 0.3 ± 0.2 | 0.54 ± 0.07 |
| R142A | 0.20 ± 0.09 | 0.35 ± 0.03 | 0.02 ± 0.02 | 0.10 ± 0.01 | 1.2 ± 0.4 | 0.70 ± 0.07 |

limiting in the catalytic cycle of WT ASNS or site-specific variants. In addition, given that the gate to the tunnel is open in the two variants, as seen in the MD simulations (Fig. 6) and in the cryo-EM structure of the unliganded R142I variant (Supplementary Figs. 29, 30), ammonia trafficking is expected to be maintained at the WT level and so have little impact on steady-state kinetic properties. Replacing the positively charged Arg-142 side chain by the smaller, hydrophobic side chains of isoleucine or alanine, however, clearly alters how the C-terminal synthetase domain in human ASNS responds to different nitrogen sources (Table 1). This difference can be explained if free ammonia directly accesses the synthetase site, as we have discussed previously[54], and ammonia translocation from the glutaminase to the synthetase site is impaired for some reason. As a result, these kinetic findings suggest that the function of Arg-142 may be more than merely a molecular gate under turnover conditions.

## MD simulations of the R142I/β-aspartyl-AMP/MgPP$_i$ ternary complex

Based on studies of carbamoyl phosphate synthetase[91], we hypothesized that the structure and/or dynamics of the tunnel in the R142I variant might be altered by the presence of the β-aspartyl-AMP intermediate. Two possibilities are that the tunnel becomes discontinuous or that ammonia, produced in the N-terminal active site, can leak into bulk solution. We therefore performed an additional set of MD simulations on the R142I/β-aspartyl-AMP/MgPP$_i$ ternary complex to explore the molecular mechanism of the nitrogen source preference seen in the R142I variant. We built a model of the cognate R142I ternary complex by replacing Arg-142 in the initial ($t = 0$ ns) structure of the human ASNS/β-aspartyl-AMP/MgPP$_i$ ternary complex by isoleucine, which was then used to generate MD trajectories under conditions identical to those described above for the WT ASNS ternary complex (Supplementary Fig. 33). Unexpectedly, the presence of the β-aspartyl-AMP intermediate in the synthetase active site significantly perturbed the structural properties of the tunnel in this ternary complex (Fig. 6b and Supplementary Fig. 33). Thus, a discontinuous tunnel was observed in the trajectory snapshots taken at 20, 80 and 140 ns, despite the Ile-142 side chain adopting a conformation allowing ammonia access. These MD-based results indicate that ammonia translocation is impaired between the active sites in the apo-R142I variant, a conclusion that is consistent with our kinetic measurements. Moreover, an additional branch of the tunnel that is seen in the final trajectory snapshot ($t = 200$ ns) for the R142I/β-aspartyl-AMP/MgPP$_i$ ternary complex may represent the development of a conduit by which ammonia can leak from the glutaminase site.

## Discussion

The successful determination of a cryo-EM structure for human ASNS establishes the physiologically relevant form of the human ASNS dimer and provides a basis for understanding how cytotoxic ASNS inhibitors, which are potential anti-cancer agents[92], might interact with the enzyme[93]. More generally, our findings demonstrate the utility of cryo-EM for identifying functionally important conformational changes[94], even at the level of a single residue in an enzyme possessing multiple active sites. Our 3DVA-derived hypothesis is that Arg-142 functions as a gate that controls the access of ammonia to the tunnel; it may also

participate in maintaining the structural integrity of the tunnel when β-aspartyl-AMP and MgPPi are present in the C-terminal active site. This gating mechanism in ASNS therefore differs from observations on other Class II amidotransferases for which tunnel formation and activation of glutaminase activity result from large conformational rearrangements that are initiated by substrate binding[95]. For example, the presence of 5′-phosphoribosyl-1′-pyrophosphate (PRPP) in the C-terminal active site is required to produce the open form of the intramolecular tunnel in *Escherichia coli* PRPP amidotransferase (Supplementary Fig. 34a)[58]. MD simulations have confirmed that PRPP binding stabilizes the conformation of a key active site loop, which is disordered in the free enzyme[96]. A large-scale rearrangement of the C-terminal tail in glutamine fructose-6-phosphate amidotransferase (GFAT) is also observed when fructose-6-phosphate binds to the synthetase site in the C-terminal domain of the enzyme[59], leading to sequestration of the substrate from water and ammonia tunnel formation. Generating a continuous, open tunnel in GFAT, however, requires reorientation of the two domains and rotation of the Trp-74 side chain, a structure that is unlikely to be adopted by the free enzyme (Supplementary Fig. 34b)[97].

Our MD simulations provide evidence that the continuous form of the intramolecular tunnel is energetically favored by formation of β-aspartyl-AMP and/or MgPPi in the C-terminal domain of the WT enzyme. More importantly, these calculations raise the possibility of a role for Arg-142 in maintaining the structural integrity of the tunnel after formation of the β-aspartyl-AMP intermediate; this finding may underpin the observed kinetic properties of the R142I variant (Table 1) although additional experiments and simulations will be needed to support this hypothesis. It is evident, however, that the motions of additional residues are involved in controlling the structural properties of the ammonia tunnel. Identifying these motions by MD simulation and validating them experimentally will yield additional insights into how continuous ammonia tunnel could be formed thereby regulating ammonia translocation in human ASNS. However, such investigations lie outside the scope of this paper and will be reported elsewhere.

In conclusion, the work outlined here illustrates the power of combining 3DVA-based observations and MD simulations to generate hypotheses about side chain function. Our strategy provides evidence for the idea that conformational changes in the Arg-142 side chain, located in the glutaminase domain, impact catalytic events in the synthetase active site at a distance of ~20 Å away. Taken overall, our results provide an example of how global regulatory mechanisms are embedded within the protein scaffold far away from the active site(s) and emphasize the need to consider the total conformational ensemble of proteins in efforts to obtain enzymes by de novo computational design[98,99].

## Methods

### Protein expression and purification

The expression and purification of WT human ASNS in Sf9 cells (Expression Systems, Davis CA) were carried out using a published protocol[60,61], except that 5 mM β-mercaptoethanol was present in all buffers used during protein purification. Briefly, a 1 L culture of Sf9 cells ($1.5 \times 10^6$ cells mL$^{-1}$) was infected with the baculovirus expressing

recombinant human ASNS or the two variants with MOI = 4.0, the infected cells were then incubated at 27 °C for 96 h before being harvested by centrifugation and frozen in liquid $N_2$. The resulting pellet was then stored at −80 °C until lysed in 50 mM Tris-HCl buffer, pH 8.0, containing 500 mM NaCl. Using a spin column, each sample of recombinant enzyme was concentrated to 9 mg/mL in 25 mM Tris-HCl, pH 8.0, containing 200 mM NaCl and 5 mM DTT. In the case of the two tunnel variants, R142A and R142I, cell pellets were sonicated for 4 min in lysis buffer containing 100 mM EPPS, pH 8, containing 300 mM NaCl, 50 mM imidazole, and 5 mM TCEP. After centrifugation at 16,743 × g for 30 min at 4 °C, the supernatant was loaded onto a Ni-NTA column equilibrated with the same buffer at 4 °C. Each protein was eluted with 100 mM EPPS, pH 8, containing 300 mM NaCl, 250 mM imidazole, and 5 mM TECP. Fractions containing protein were collected, and exchanged into 100 mM EPPS, pH 8, containing 150 mM NaCl, and 5 mM TCEP, using a PD-10 size exclusion column. The purified enzyme was concentrated using a spin filter, and its concentration was determined by Bradford assay. Aliquoted ASNS variant samples were stored at −80 °C. For cryo-EM work, the sample buffer containing the ASNS R142I variant was exchanged to 25 mM Tris-HCl, pH 8.0, containing 200 mM NaCl and 5 mM DTT by dialysis. The dialysate was concentrated by a spin column to 3.2 mg/mL.

### Cryo-EM sample preparation and data collection

300 mesh UltrAuFoil R1.2/1.3 grids were glow-discharged for 1 min with a current of 15 mA in a PELCO easiGlow system before being mounted onto a Mark IV Vitrobot (FEI/Thermo Fisher Scientific). The sample chamber on the Vitrobot was kept at 4 °C with a relative humidity of 100%. A 3.0 μL sample of either recombinant WT ASNS or the R142I variant, at concentrations of 9.0 mg/mL and 3.2 mg/mL, respectively, was applied to the grid, which was then blotted from both sides for 4 s with the blot force set at 0. After blotting, the grid was rapidly plunge-frozen into a liquid ethane bath cooled by liquid nitrogen.

Single particle data was collected by a 200 kV Glacios transmission electron microscope (FEI/ThermoFisher Scientific) equipped with a Falcon4 camera operated in Electron-Event Representation (EER) mode[100]. Movies were collected using EPU at 150,000x magnification (physical pixel size 0.93 Å) over a defocus range of −0.4 to −2.0 μm and a total accumulated dose of 40 e/Å². Camera gain reference was taken at the end of the run. A total of 3584 movies were acquired for WT ASNS and 4,101 movies for R142I variant. Data were processed using cryoSPARC v3.2.2 and v4.4.1 over the duration of the project. Each movie was split into 40 fractions during motion correction with an EER upsampling factor of 1. After motion correction and CTF estimation, micrographs were manually curated based on relative ice thickness and CTF resolution fit such that 2103 micrographs out of corresponding 3584 movies for WT and 2649 out of 4101 movies for the R142I variant were chosen and used for subsequent data processing.

### Assessing the mode of ASNS dimerization and structure determination

To gauge how ASNS dimerizes in an unbiased fashion, particles were initially selected using the blob pick option in cryoSPARC[62]. 2D class averages showed what appeared to be 2D projections of a human ASNS dimer similar to that seen in the X-ray structure[49] (Supplementary Fig. 2b). Further, 769,800 particles were subjected to two rounds of ab initio reconstruction, resulting in a total of 12 initial reconstructions (Supplementary Fig. 2d). One reconstruction clearly resembles the head-to-head dimer form of ASNS (compare Supplementary Figs. 2d, e), and the rests exhibited small densities. In light of this observation, particle selection was then carried out by template-based picking, using a template generated from the X-ray structure of DON-modified human ASNS (6GQ3) in UCSF Chimera[101]. The full cryo-EM data processing workflow of WT ASNS is illustrated in Supplementary Fig S3. Bad particles were removed using several rounds of 2D

classification followed by 5 rounds of ab initio reconstructions, and successive heterologous refinement, non-uniform refinement with CTF and defocus correction, and the imposition of $C_2$ symmetry gave rise to a map with 3.5 Å overall resolution. The map was sharpened using the program DeepEMhancer[102] as implemented on the COSMIC² site[103]. Model building started with fitting the crystal structure of DON-modified human ASNS (6GQ3)[49] into the EM map using rigid body refinement by REFMAC[104]. The resulting model was then iteratively refined by real-space refinement in Phenix[105] followed by manual inspection and refinement in Coot[106]. Figures for the cryo-EM density maps and models were generated with Chimera[100] or ChimeraX[107]. The full cryo-EM data processing workflow of R142I variant is illustrated in Supplementary Fig S25. The data processing for ASNS R142I was carried out by a template-picking strategy. Bad particles were removed using several rounds of 2D classification followed by 5 rounds of ab initio reconstructions, and successive heterologous refinement, non-uniform refinement with CTF and defocus correction, and the imposition of $C_2$ symmetry gave rise to a map with 3.35 Å overall resolution. The map was sharpened using the program DeepEMhancer[102] as implemented on the COSMIC² site[103]. Model building for the R142I ASNS variant was carried out by essentially the same procedure as that described for WT ASNS except that the EM structure of apo-ASNS (8SUE) was used as the initial model.

### 3D variability analysis (3DVA) of the EM map for human ASNS

Each EM map of WT ASNS and the R142I variant was separately subjected to 3D variability analysis (3DVA) in cryoSPARC v4.4.1 filtered at 4 Å resolution with the mask previously used in a final non-uniform refinement[62]. Five principal components were used in both analyses. A movie for each component was made with UCSF Chimera[100] using frames generated by the 3DVA display program in a simple mode in cryoSPARC v4.4.1 (Supplementary Movies 1−5 for WT apo-ASNS, Supplementary Movies 6−10 for the R142I variant; details are provided in Supplementary Table 6). Movie frames were subjected to variability refinement in Phenix, using 50 models per map, yielding a series of 20 models per component. Backbone RMS fluctuations for each residue in the set of 100 structures obtained for WT ASNS were then computed using the CPPTRAJ software package[84].

### Molecular dynamics simulations

The ROBETTA server[81] was used to build a model of full-length human WT ASNS in order to add the residues that were not observed in the X-ray structure. In addition, this structure contained an unmodified side chain for Cys-1. The protonation state of ionizable residues at pH 7.4 was determined by calculation using the PROPKA algorithm[108], as implemented in Maestro (Schrodinger Suite 2018-01) (Supplementary Table 7).

A model of the human ASNS/β-aspartyl-AMP/MgPP$_i$ ternary complex was then obtained by positioning the Mg(II) salt of inorganic pyrophosphate (PPi) into the initial WT apo-ASNS model such that MgPPi was in an identical location to that observed in GMP synthetase[109] and other members of the PP-loop ATP pyrophosphatase superfamily[110]. β-aspartyl-AMP was docked into the model of the binary complex so that the nucleobase was in a similar site to that seen for AMP in the X-ray crystal structure of the bacterial enzyme[50]. The protonation state of ionizable groups in the protein and the two ligands were assigned using PROPKA[108] and Epik[111], respectively. As a result, the amino and carboxylate groups of the aspartate-derived portion of β-aspartyl-AMP formed complementary electrostatic and hydrogen bonding interactions with Gly-365, Asp-367, Glu-368 and Arg-403; these four residues are conserved in known glutamine-dependent asparagine synthetases (Supplementary Fig. 11).

Each of these models was energy minimized before being placed in a box of TIP3P water molecules[112], which was sized to ensure a minimum distance of 10 Å between protein atoms and the sides of the

box. After the addition of sodium ions to neutralize the total charge of the system, each model was heated to 300 K and equilibrated in the NPT ensemble ($P = 101,325$ Pa). Four 200 ns NPT MD simulations for each of these two models were then carried out, with the OPLS_2005 force field[113] parameters describing the protein and the β-aspartyl-AMP intermediate (Supplementary Data 1). In a control simulation, a microsecond trajectory was computed for the apo-ASNS monomer under identical conditions. These calculations were performed using the DESMOND MD package (D.E. Shaw Research, New York, NY, 2018). Snapshots were taken at 200 ps intervals in MD simulations of the trajectories for WT ASNS and the WT ASNS/β-aspartyl-AMP/MgPP$_i$ ternary complex (Supplementary Movies 11, 12, respectively).

Each of the two, solvated equilibrated models (WT ASNS and the WT ASNS/β-aspartyl-AMP/MgPP$_i$ ternary complex) were also submitted to well-tempered metadynamics (MTD) simulations[85,86], as implemented in the DESMOND software package until convergence was achieved at 1400 ns for apo-ASNS or at 600 ns for the ternary complex (see below). Two distance-based collective variables were defined for these simulations (Supplementary Figs. S15, S21), corresponding to the distances between the center-of-mass (COM) of the guanidinium group in Arg-142 and the COM of the carboxylate moieties in either Asp-405 (CV1) or Glu-414 (CV2). As outlined above, protein residues and the β-aspartyl-AMP and MgPP$_i$ ligands were described by the OPLS-2005 force field and the TIP3P potential was used for the water molecules. In both WT-MTD simulations, Gaussian widths for CV1 and CV2 were set to 0.2 Å and 0.3 Å, respectively, with the initial height of the Gaussian being 0.3 kcal/mol. Gaussians were deposited at 1 ps intervals and the bias factor (defined as RΔT in DESMOND) for Gaussian rescaling was 1.7 kcal/mol. Trajectory snapshots were recorded at 10 ps intervals. Convergence in both of these MTD simulations was confirmed (Supplementary Figs. S35, S36) using a script developed for analyzing DESMOND-derived trajectories (Schrodinger, Inc.), which we include elsewhere this paper (Supplementary Data 2).

A computational model of the WT ASNS dimer was built by superimposing two copies of the initial apo-ASNS monomer model on the X-ray crystal structure of human ASNS.[49] As outlined above for the MD simulations of the apo-ASNS monomer, the resulting dimer was energy minimized, placed in a box of explicit TIP3P water molecules, and heated/equilibrated to 300 K. A 200 ns NPT MD simulation of the resulting system was then carried out, with the OPLS_2005 force field[113] parameters describing the protein. Snapshots were taken at 200 ps intervals in MD simulations of the trajectory. No significant differences in the RMS fluctuations of residues are seen, except in the modeled C-terminal region, when monomer A and monomer B of the ASNS dimer are compared to each other (Supplementary Fig. 37). Based on these findings, the use of monomeric enzyme structures, for computational convenience, in all production MD simulations of human apo-ASNS, the human ASNS R142I variant, and the WT ASNS/β-aspartyl-AMP/MgPP$_i$ and R142I/β-aspartyl-AMP/MgPP$_i$ ternary complexes appeared to be justified.

Finally, computational models of the apo-R142I ASNS variant and the R142I/β-aspartyl-AMP/MgPP$_i$ ternary complex were built by graphically replacing Arg-142 by isoleucine (Maestro GUI) in the initial, unoptimized models of the apo-ASNS and the ASNS/β-aspartyl-AMP/MgPP$_i$ ternary complex. These R142I-containing models were energy minimized and heated/equilibrated following identical procedures to those outlined above. Once again, four 200 ns NPT MD simulations were carried out with the DESMOND MD package using identical conditions to those outlined above for the WT enzyme and its ternary complex. Snapshots were taken at 200 ps intervals in MD simulations of the trajectories for the R142I variant and the R142I/β-aspartyl-AMP/MgPP$_i$ ternary complex (Supplementary Movies 13, 14, respectively).

All MD simulations employed periodic boundary conditions (cubic box), and long-range electrostatic interactions were calculated by particle mesh Ewald[114] with short-range interactions being truncated at 9 Å (Supplementary Data 3 and Source Data). Trajectories were analyzed using CAVER 3.0 (default settings) to determine the conformation of the intramolecular tunnel in sampled protein structures[89] and backbone RMS fluctuations for the structures in the four MD trajectories were computed using CPPTRAJ[83].

## Steady-state kinetic measurements

The rate of inorganic pyrophosphate (PPi) production catalyzed by recombinant, WT human ASNS and two ammonia tunnel variants was determined in a continuous assay format using the EnzChek™ Pyrophosphate Assay Kit (Molecular Probes, Eugene, OR). Briefly, ASNS-catalyzed pyrophosphate production was measured at 37 °C over a 4 min period by monitoring the absorption at 360 nm. In this assay, the reaction (400 μL) was initiated by the addition of enzyme (50 nM) to a reaction mixture containing 2-amino-6-mercapto-7-methylpurine ribonucleoside (0.2 mM), purine nucleoside phosphorylase (1 U/mL), inorganic phosphatase (0.09 U/mL), and substrates in 100 mM EPPS, pH 8, containing 2 mM dithiothreitol, 10 mM MgCl$_2$ and 5 mM ATP. When L-aspartate was varied (0–12.5 mM), the concentration of either L-glutamine (20 mM) or NH$_4$Cl (100 mM) was fixed. Similarly, when either glutamine (0–20 mM) or NH$_4$Cl (0–100 mM) was varied, the L-aspartate (10 mM) concentration was constant. The concentration of free ammonia in the assay was calculated using the reported pKa value at 37 °C[115]. All assays were performed in duplicate, and data analysis was performed using GraphPad Prism (GraphPad Holdings, La Jolla). Source data are provided as a Source Data file.

## Reporting summary

Further information on research design is available in the Nature Portfolio Reporting Summary linked to this article.

## Data availability

The primary data supporting the results of this study are available within the paper and its Supplementary Information. EM-derived structures reported in this study have been deposited in both the PDB under accession codes 8SUE (WT human ASNS) and 9B6G (R142I ASNS variant). These structures have also been deposited in the Electron Microscopy Data Bank as EMD-40764 (WT human ASNS) and EMD-44253 (R142I ASNS variant). Supplementary Movies 1–14 (Supplementary Table 6), additional EM-associated data, 3DVA-associated data, and MD and metadynamics trajectories are available in the Zenodo database (Supplementary Table 8). Source data are provided with this paper.

## Code availability

The script used to assess convergence of the MTD simulations is included in the Supplementary Materials for this paper (Supplementary Data 2).

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

## Acknowledgements
We thank Francesco Ortuso for technical support of the computational studies, and Davide Branduardi for his help in developing the script for analyzing the convergence of the MTD simulations. The authors would also like to acknowledge the Indiana University School of Medicine EM facility and NIH/NIGMS, S10 OD028723 for supporting this work. Additional funds for the EM structure determinations were provided by the Indiana University School of Medicine (Y.T.), and, in part, NIGMS award number R01GM111695 (Y.T.) and the American Cancer Society award number DBG-23-1038947-01-IBCD (Y.T.). This work was also supported by the CNRS (V.C.), and the French National Research Agency grant number ANR-19-CE11-0023-01 (V.C.). Funding for the computational studies was provided by the Biotechnology and Biological Sciences Research Council, grant number P/018017/1 (N.G.J.R.), and by the Italian Ministry of University and Research (MUR) grant PRIN 201744BN5T (S.A.). Kinetic characterizations of WT ASNS and the R142A and R142I ASNS variants were made possible by start-up funding from the Florida State University (W.Z.). C.G. was supported by funds from the EU, project number FSE-FESR PON-RI 2014-2020. The content is solely the responsibility of the authors and does not necessarily represent the official views of the Indiana University School of Medicine. The European Committee and Regione Calabria also decline any responsibility concerning the use of the information disclosed in the paper.

## Author contributions
A.C., W.Z., Y.T. and N.G.J.R. designed the research. Y.T. and M.V. collected data, and Y.T. solved the EM structures. V.C. and Y.T. carried out the 3D variability analysis. A.J.N. performed the kinetic assays under the direction of W.Z. A.C., A.L. and C.G. performed the computer simulations under the guidance of S.A. and N.G.J.R. MD simulation data were analyzed by A.C., C.G., W.Z., S.A. and N.G.J.R. Finally, the paper was co-written by W.Z., Y.T. and N.G.J.R. with input from all other authors.

## Competing interests
The authors declare no competing interests.
