## [Transparent Peer Review file · Nature Communications]

3D Variability Analysis Reveals a Hidden Conformational Change Controlling Ammonia Transport in Human Asparagine Synthetase

Corresponding Author: Professor Nigel Richards

Version 0:

Reviewer comments:

Reviewer #1

(Remarks to the Author)

Summary: In this study, researchers used 3D variability analysis (3DVA) combined with atomistic molecular dynamics (MD) simulations to investigate the dynamic motions of human asparagine synthetase (ASNS). By solving the structure of ASNS and performing 3DVA, they suggest that a single side chain's dynamic motion (Arg142) regulates the interconversion between open and closed forms of an intramolecular tunnel. The opening of this tunnel allows for the translocation of ammonia, which is necessary for ASNS's catalytic function. MD followed up on this initial finding to determine exactly how

The study highlights the power of cryo-EM in identifying localized conformational changes and demonstrates how conformational dynamics can regulate the function of metabolic enzymes with multiple active sites. However, the lack of experimental electron density shown in the figures (or available publicly) makes it difficult to assess the claims in this study. Additional forward tests of the importance of this blockage via mutagenesis may also uncover why it has to be regulated at all. If this is out of the scope of the current paper, it should be hypothesized and speculated upon in the discussion.

Major Points:

1. In your figures, Please show electron density and all individual atomic positions. This includes Fig. 1d, 2a, and all of Figure 3.
2. Please show the PCA of the 3DVA. Clarify whether this was done on the entire structure or the tunneling residues. If done on the entire protein, please comment and show if other changes were seen elsewhere.
3. In the RMSD analysis, please clarify what EM coordinates you are using. Are you comparing all structures from the 3DVA? Please also provide raw values as well as normalized values.
4. Your results do not support the claim 'Our results suggest that changes in the C-terminal active site are propagated over a distance of approximately 20 Å, leading to tunnel opening and ammonia translocation'. While the data here shows that the tunnel can move between an open and closed state in apo form as part of the native fluctuations (revealed by the PCA analysis). There is no information presented on how this information is propagated nor how the active site or binding interacts with this motion. Please change the wording of this or explain the mechanism.

Minor Points:

1. Neither the PDB nor the map is publicly available, making it difficult to examine the structures and map independently. Please release them. Also, include information and metrics regarding map sharpening and map-to-model fit. Zenodo is a good option for the 100 structures from the PCA analysis.
2. In Figure 1d, please label the amino acids and chains. Please provide experimental density corresponding to the positions of these residues in this figure or a supplementary figure.
3. In Figure 2a, please provide a legend for what each color represents.
4. How did you determine 5 PCAs for the 3DVA analysis?
5. Please provide details on the normalized RMSF. How was this normalization done?

- In Figure 4, please provide a legend for all colors of amino acids and tunneling.
- Please deposit the coordinate files for the 100 structures used in the 3DVA study and (ideally also two MD-derived trajectories) on Zenodo or similar repository.

Stephanie Wankowicz and James Fraser

Reviewer #2

(Remarks to the Author)

The manuscript reports cryo-EM model and associated 3DVA and MD simulation analysis of human asparagine synthetase. Beyond some verifications between MD and cryo-EM, a key result is the observation of closed confirmation in the 3DVA analysis even when the ligand is absent. My comments are limited to the MD simulation part of the work.

First, the idea of a protein conformational pre-equilibrium prior to substrate binding is already very well established. So it does not strike as a surprise unless the functional relevance for the synthetase is etched out.

Second, the MD simulation is weak in that only one part of the transition is visible, which statistically means very little. Unless in simulations the entire 'open to close back to open' conformations is observed multiple times, the mentioned dynamic pathway of closing have minimal relevance. Extended sampling tools like CryoFold (Matter 2021 by Shekhar et al) are meant for this purpose. The movements from 3DVA similarly also has to be verified by tools like manifoldEM or Polaris to have any interpretation of the dynamics in terms of the energy landscape.

Third, so much is already known from crystallography that the cryo-EM piece looks incremental at best. This can be a presentational issue, but dampens my excitement about the findings.

Finally it was already shown by Schulten and co-workers that the fluctuations in MD and those from the maps when comparable can offer a reasonable map-modeling criteria. The work should be acknowledged and indeed offers support to the article.

Reviewer #3

(Remarks to the Author)

Review of manuscript ""

The authors use 3D variability analysis (3DVA) of an EM structure for human asparagine synthetase (ASNS) in combination with atomistic molecular dynamics (MD) simulations to detail how dynamic motions of a single side chain mediates interconversion of the open and closed forms of a catalytically relevant intramolecular tunnel. This interconversion between open and closed states allows ASNS regulating catalytic function. The findings demonstrate the utility of cryo-EM in providing information on protein motions, even at the level of a single residue in an enzyme possessing multiple active sites. This is excellent work which is presented in a clear and distilled manner and in my opinion meets the high requirements of Nat. Commun. However, the authors should carefully consider the suggestions below. Especially the point regarding performing additional replica is crucial.

1. Abstract:

- Sentence logic unclear: "How dynamical motions in enzymes might be linked to catalytic function is of significant general interest, although almost all relevant experimental data, to date, has been obtained for enzymes with a single active site."
- "dynamic motions": Isn't all motion dynamic? Remove redundancy.

c.

2. Introduction:

- "Moreover, experimental evidence has been published that supports the existence of thermal networks in enzymes, which mediate energy transfer between the active site and external solvent.". I don't think this statement is supported by actual evidence in the cited articles. This is a proposal, which is lacking any real hard evidence. Commonly, we assume equipartition of kinetic energy in classical system, like heavy atoms in proteins. I suggest modifying the sentence to reflect better what has been actually proven scientifically.

3. Results:

- The residues involved in interactions stabilizing the closed and open forms must be included and marked in Fig. 3, 4.
- MD simulations: The events seen in the MD simulations could be happening by chance, as this is a deterministic method. The authors must perform an additional 2-4 MD simulations with different initial conditions (e.g., velocities) of both systems simulated to obtain statistically significant data, to reach 3-5 replicas that is the accepted number for vanilla MD simulations. See f. ex. Knapp, B., L. Ospina, and C.M. Deane, Avoiding False Positive Conclusions in Molecular Simulation: The Importance of Replicas. Journal of Chemical Theory and Computation, 2018. 14(12): p. 6127-6138.
- It would also be good to extend the length of the simulations to verify that the conformational transitions are reversible on the time scale of the simulations.
- Why was diffusion of NH3 not modeled? This is fairly straightforward.

4. Methods:

- "The hydrogen bonding networks in each of the models were then optimized, and the protonation state of ionizable residues at pH 7.4 determined by calculation." The type of calculation used must be specified.

Reviewer #4

(Remarks to the Author)

The manuscript by Richards and colleagues combines cryo EM studies with MD simulations to obtain insights into dynamic processes that regulate the activity of the Human Asparagine Synthetase enzyme (ASNS).

As the authors write, it is highly important to understand how motions in enzymes are correlated with function. Modern methods in structural biology have provided a very large number of static images of bio-molecular complexes. These static images are important, but fail to provide information on how structural changes in these complexes regulate function. In that light, the current study is an important example that links the motion of a single side-chain with ammonia transfer from one active site in ASNS to the other one.

Despite this importance, I have a number of major remarks to the manuscript. In many cases I am not sure if the conclusions are sufficiently covered by the (limited) data. To fully claim the proposed model, additional cryo EM structures, e.g. in the presence of substrates, would be helpful/ needed.

1. The authors mention at multiple locations that NMR spectroscopy is limited. I do agree with that to some degree, but recent advances have made NMR amenable to study enzymes with multiple active sites (e.g. <https://doi.org/10.1038/s41467-021-22968-6>) or of very large complexes (e.g. <https://doi.org/10.1126/science.1233066>, <https://doi.org/10.1038/s41589-022-01111-6> or <https://doi.org/10.1038/nature12581>) to mention a few random examples.
2. Did the authors try to do a template-based particle picking with the bacterial homolog (AS-B)? This is important in case the authors would like to claim that other dimers are not present.
3. Is it true that only 13.2% of the particles are the dimeric enzyme. If yes, does that not imply that the enzyme exist in a monomer ↔ dimer equilibrium. Is that also observed in solution (is there SAXS or SEC data)? Is the dimerization relevant for catalysis? This should be clearly stated, as currently I am not sure what the consequence of the (kind of) dimerization is. Does the “monomer” have the same R 142 behavior regarding the tunnel?
3. “it is likely that these regions are disordered rather than being absent due to proteolysis.” I agree that regions that are not well visible are likely not well ordered. Is there any indication of proteolysis (e.g. from an SDS gel).
4. “especially because this residue, Arg-142, is conserved in mammalian asparagine synthetases” This sounds more impressive than it is, as almost all residues are conserved in mammals according to figure S8. As the authors claim an important role for R142 in the regulation of the access to the ammonia tunnel the question arises how this important aspect takes place in the bacterial enzymes (that don't have the Arg in that position). How does the Arg (mammalian) to Ile (bacterial) mutation in the human enzyme effect activity/ regulation?
5. There is some density shown in figure S5. I would appreciate a clear view of R142 in addition.
6. “In contrast, the intramolecular tunnel is continuous when glutamine and AMP are bound within the active sites of the C1A AS-B variant (i.e., it is in an “open” form).” The authors compare two different things here: the apo human structure (closed), with the E. coli in the presence of AMP and glutamine. It is thus not clear if the AMP binding caused the opening of the tunnel (as the authors suggest) or if the tunnel opening is due to the glutamine binding or the differences in amino acid sequence. One should really compare the tunnel opening of the human protein in the presence and absence of (one or two) ligands, preferably by cryo-EM.
6. The 3DVA analysis is impressive, but I would need to see densities that would allow for the unambiguous placement of the R 142 side chain. At the resolution of the structures I am not sure if it is possible to place R 142 accurately. What percentage of the complex is open/ closed?
7. The 3DVA analysis suggests that the apo protein (that the authors study here) is partially in an “tunnel open” state. That makes little sense to me, as then ammonia could “escape”. In case the “plug” is only closed part of the time the complete regulation is non-functional.
8. Is the MD done on a monomeric form of the enzyme? That appears odd, as the authors clearly show a dimeric form in the cryo-EM.
9. To me there are no significant differences in the tunnel opening between the apo and ternary complex. The MD simulation is too short to draw conclusions regarding the statistics of tunnel opening/ closing. In Fig. 4C the tunnel appear actually wider in the apo state. In this figure the two active sites as well as the start and end of the tunnel should be clearly indicated.
10. The link between the substrate binding and tunnel opening is described in the last part of the results, but this is not clearly shown in a figure. How does this “look”. Also, the open state should be structurally compared to the open state in the bacterial enzyme.
11. The importance of R 142 in channel closing should be shown experimentally, e.g. by performing activity assays that

assess if the glutaminase activity is larger when the tunnel is open (R to A mutation?).

12. "The role of Arg-142 in controlling conformational plasticity in the intramolecular tunnel of apo-ASNS contrasts sharply with observations on other Class II amidotransferases in which substrate binding is essential for tunnel formation and activation of glutaminase activity". Please explain this in detail, as also in the present case substrate binding seems to be important in opening the channel. And how do the authors know that binding of additional substrates has no influence on the channel, there is no EM or MD data regarding this.

13. Are any domain motions or re-orientations observed in the MD, as has been observed for related complexes?

14. "Our results suggest that changes in the C-terminal active site are propagated over a distance of approximately 20 Å,". I don't see this. The authors show some very local changes only, e.g. in Fig. 4C.

Version 1:

Reviewer comments:

Reviewer #1

(Remarks to the Author)

Dropbox is not a good solution for storing the data as they have done. Suggest using zenodo or dryad.

Reviewer #2

(Remarks to the Author)

Even a long molecular dynamics simulation might not capture the transition with complete statistical significance. Normally free energy simulations are important to ensure statistical significance of a movement and qualify it as biophysically relevant dynamics.

The new analysis definitely strengthens the article and suggests that the movements seen in 3DVA might have some physical relevance as seen in MD. But unless a free energy analysis of the MD simulations is performed to indeed show that the energy changes (including barriers) associated with the conformational are thermally accessible, the data will remain weak.

Reviewer #4

(Remarks to the Author)

The authors have made significant efforts to improve the manuscript and have now presented a very convincing version where my initial criticism has been fully resolved. I especially appreciate the activity assays that have been included and that confirm the computational work. Note that I really appreciate the computational work, but that I am not an MD expert and can thus not 100% ensure that all computational details are fully correct. In case the other reviewers that might be experts in MD agree I support publication of the manuscript without further edits.

Reviewer #5

(Remarks to the Author)

In this paper, Coricello and coworkers report cryo-EM and MD simulations of Human Asparagine Synthetase and identify Arg-142 as key residue for regulating the ammonia tunnel. Considering that some X-ray data was already available, and that the new cryo-EM and MD data are not providing enough evidence of the reported claims, the paper lacks the novelty required for publication in Nat Commun. Additionally and more importantly, the conclusions withdrawn are not well supported by the MD data, and the analysis of the simulations is far from optimal.

The following major comments should be addressed:

Introduction:

It is really surprising that a paper focused on the analysis of conformational dynamics by means of cryo-EM and MD contains an introduction with only a few citations to computational papers covering the analysis of the conformational landscapes. For instance:

"Certainly, allosteric regulation of enzyme activity depends on correlated residue movements, and computer-based simulations ^{***}(two citations added: one experimental. The other computational)^{**} support the idea that altered dynamics can be correlated with improvements in the catalytic efficiency of "designer" enzymes as they undergo optimization by directed evolution ^{**}(again experimental papers are cited (99 Arnold, 21 lane)^{**}. A long list of more recent publications performing MD simulations could be instead cited, to name a few: reference 49 of the paper, 10.1021/jacs.0c04924, 10.1039/C8CC02426J.

"Much less is known about how dynamical motions impact catalysis in enzymes that possess two, or more, active sites, such as glutamine-dependent amidotransferases or tryptophan synthase." It is surprising again that no MD simulations are cited,

when many papers have been focused on analyzing enzymes with multiple active sites, such as ImGPs, TrpS: 10.1038/s41467-023-37956-1, 10.1021/jacs.9b03646, 10.1016/j.bpj.2021.11.2888

Results:

The authors analyze the effect of Arg-142 conformation as defined as open/closed (or in some cases partially open), on both the X-ray data and some snapshots along the MD simulations performed. How is this open/closed defined? Based on a distance, angle, dihedral of the Arg-142 sidechain? How do they know that Arg-142 conformation is directly related to tunnel formation? In the included Figures (4, 6) Arg-142 is not displayed, so such correlation cannot be extracted. It is therefore not clear how the authors know that Arg "open/closed" regulates tunnel formation.

It is stated in the text: "Although no change is seen in two PCs (components 0 and 2) (Figs. 3b and 3d), the Arg-142 side chain is re-positioned in two other PCs to yield an "open" form of the tunnel." Which are the main coordinates contributing to the different PCs? Only the tunnel residues are displayed, but many other changes might be observed in the protein structure as shown in the attached movies. It is also stated that the PCA is done considering the coordinates of the entire enzyme in the dimeric form, so the contribution of the Arg-142 conformational change might be rather low in all described PCs.

The authors mention at the end of the results section: "This level of agreement between experiment and theory strongly supports the EM-derived mechanism in which movement of the Arg-142 side chain results in tunnel opening and ammonia translocation." In our opinion, such statement cannot be made as the correlation between Arg-142 conformation and tunnel formation has not been demonstrated.

Regarding the model generated for the MD simulations, again we have several concerns:

"Using the Robetta server, we constructed a model of full-length human apo-ASNS". It is not clear to us why the more recent and accurate AF2 multimer tool was not used to reconstruct the full model. How reliable is this Robetta structure?

"The resulting model, which was independent of the cryo-EM structure". It is not clear what the authors mean with this statement. How reliable is the model as compared to cryo-EM structures?

The authors performed MD simulations on both the monomeric and dimeric forms of the enzymes. However, what is the oligomerization state of the enzyme in solution? This could be determined experimentally, which will help then deciding which model is best to analyze and include in the main text.

In Figure S18, a comparison of backbone RMSF values computed from 200 ns MD trajectories of the full-length apo-ASNS monomer and the apo-ASNS dimer is displayed. Although the authors mention that no significant differences were observed between monomer and dimer, the RMSF displayed in SI does show significant differences: the dimer appears to be much more flexible than the monomer, especially in the 480-540 region for monomer B.

In Figure 5, large differences between the cryo-EM structures and MD data are also observed in the region around 475-500, and 200-230. Why is that? Where are the regions located and how important are for the catalysis? The same is observed when comparing apo and ternary complex MD simulations, thus indicating that these regions play a key role for binding and catalysis. The authors mention that the large deviations correspond to regions not solved in the X-ray structure, so again we wonder about the reliability of the generated model.

The accuracy of the MD simulations of the ternary complex is also not clear. As shown in Figure S29, the Ppi and beta-aspartyl-AMP are displaced along the 200 ns MD simulation. In fact one might expect the phosphate group to be strongly bound to the active site, but instead it is substantially displaced. An additional point is that focusing the analysis on the first and last frame of the MD data is not optimal, how stable are the $t=0$ and $t=200$ ns snapshots? Why these should be taken as reference?

Regarding the analysis of the network of interactions involving mostly Arg-142:

Figure S16 shows the plot of the distances between Arg-142/Glu-414 (a), and Arg-142/Asp-405. In none of the replicas, a short distance between Arg-142/Glu-414 is shown: in the displayed snapshot a distance of 2.80 Å is highlighted, however, all replicas present a very long distance of ca. 8 Å in all cases. Similarly, the distance between Arg-142/Asp-405 is maintained around 3 Å in only two of the four replicas. This is indicating that in two replicas of the MD, Arg-142 is neither interacting with Glu-414 nor Asp-405. Which interactions is therefore Arg-142 making?

"Forming the Arg-142/Asp-405 salt bridge requires breaking a network of intramolecular interactions, involving Arg-142, Glu-414, Arg-415 and Asn-75, that is present in the initial ($t = 0$ ns) structure. Given that the Asn-75 side chain forms the oxyanion hole required for glutamine hydrolysis, breaking this network provides a potential molecular mechanism by which tunnel opening might be coordinated with ammonia release in the glutaminase active site." None of these additional interactions

are displayed in any of the figures in the main text and SI, which only show the Arg-142/Asp-405 salt bridge. It is therefore not clear how the authors conclude such statement.

Regarding the analysis of the tunnels:

The comparison of the tunnels between WT and variants is also not conclusive. In Figure 6b, the tunnel for the R142I variant is displayed, which shows that at the 20 and 140 ns snapshots there is actually no tunnel. They claim that the conformation of Ile is not regulating tunnel formation, so clearly the formation of the tunnel is affected by additional residues not considered in the manuscript. The analysis of the bottleneck of the tunnel and which residues are located close to this narrowest region might be helpful to provide additional insights.

In Figure S19, a comparison of the open tunnel seen in the X-ray structure of AS-B (wheat) and the open tunnel of human apo-ASNS predicted by the MD simulations (grey) is displayed. Although the authors claim that the tunnels are similar, this is hard to tell with what is displayed in Figure S19.

The kinetic characterization of the WT and variants is not conclusive either:

“One might expect no impact on catalytic activity under our assay conditions because ammonia translocation is unlikely to be the rate-limiting step in the catalytic cycle.” Which is the rate-limiting step along the cycle? How do the authors know that there is no change in the rate-limiting step after mutation, according to their simulations substantial differences in tunnel formation are observed between WT and variant.

“Whatever the molecular basis for the altered steady-state kinetic parameters of the two ASNS variants, however, these kinetic studies support the hypothesis that Arg-142 is important for tunnel function, L-glutamine recognition, and the structural integrity of the tunnel given that free ammonia likely accesses the synthetase site directly rather than passing through the intramolecular tunnel.” What is the molecular basis of the effect of the mutation? It is not clear in their interpretation of the kinetic data.

Discussion:

“This gating mechanism contrasts sharply with structural observations on other Class II amidotransferases for which substrate binding initiates large conformational rearrangements that lead to tunnel formation and activation of glutaminase activity. [...] MD simulations have confirmed that PRPP binding stabilizes the conformation of a key active site loop, which is disordered in the free enzyme.” This is what RMSFs plots seem to indicate in the case of ASNS reported here, so we disagree with this statement.

Minor points:

In Figure 6c, it is stated that a slate color is used for $t=0$ and $t=200$ ns. This should be corrected.

In all figures where a selected distance is displayed along the MD simulation time, the same color (or at least similar color) should be used for each replica (i.e., replica 1 always displayed using green color, replica 2 red). In this way one can easily capture if a given conformational change observed for a given distance is correlated to another one in the same replica.

Reorder all figures in SI as they are cited in the main text.

Indicate the exact atoms that are used to compute the distances

In Figure S20 and 28, it is stated that the distance between Arg-142 and Glu-414 is displayed for the R142I mutant, so this should be corrected.

Reviewer #6

(Remarks to the Author)

Version 2:

Reviewer comments:

Reviewer #5

(Remarks to the Author)

The authors have substantially revised and modified the manuscript following the referees' concerns. However, many of the raised concerns have not been properly addressed, and therefore we cannot recommend publication of the paper unless the following comments are considered.

Q1: It is really surprising that a paper focused on the analysis of conformational dynamics by means of cryo-EM and MD contains an introduction with only a few citations to computational papers covering the analysis of the conformational landscapes.

A1: Almost all of these papers focus on allosteric networks, which generally impact substrate binding (K_m) rather than catalysis (k_{cat}). We agree, however, that these papers should be included and have revised the introductory text as requested by the reviewer. As a result, we have also added new citations (8, 13, 14, and 31-35).

Please note that one of the included references focuses on ImGPS enzyme, in which allostery mostly affects k_{cat} rather than K_m .

Q2: The authors analyze the effect of Arg-142 conformation as defined as open/closed (or in some cases partially open), on both the X-ray data and some snapshots along the MD simulations performed. How is this open/closed defined? Based on a distance, angle, dihedral of the Arg-142 sidechain? How do they know that Arg-142 conformation is directly related to tunnel formation? In the included Figures (4, 6) Arg-142 is not displayed, so such correlation cannot be extracted. It is therefore not clear how the authors know that Arg "open/closed" regulates tunnel formation.

A2: The definition of open and closed is based on whether the side chain blocks access to the tunnel (closed) or not (open), as seen in the EM structure. The key idea here is that Arg-142 gates the access of ammonia to the tunnel, whether the latter is in a continuous or discontinuous form. We have attempted to make this point clearer by using more precise language in the revised manuscript. We now also indicate the location of either the Arg-142 or Ile-142 side chain using red dots in Figures 4 and 6. A quantitative description of the two conformations is also now presented in Supplementary Figure S15, which forms the basis of our choice for the two collective variables used in the new metadynamics simulations (Supplementary Figure S21).

The mean and standard deviations of the set of collective variables identified and represented in Figure S15 (and subsequent ones, especially in Figure S21) should be included. What remains to be shown is that tunnel formation is regulated by Arg-142, which are the residues that form the bottleneck area of the computed CAVER tunnels? In those snapshots shown in Figures 4 and 6, the values of distances of Arg-142 and surrounding residues should be included, especially those distances used for the metadynamics simulations. I am also curious about the selection of the two CVs for performing the metadynamics simulations, both CVs are based on the distance between Arg-142 and either Glu-414 or Asp-405, so they are dependent on each other. More details should be provided on the convergency of metadynamics simulations.

Q5: "Using the Robetta server, we constructed a model of full-length human apo-ASNS". It is not clear to us why the more recent and accurate AF2 multimer tool was not used to reconstruct the full model. How reliable is this Robetta structure?

A5: AF2 was not available when we performed the original simulations and we sought to be consistent in subsequent calculations used to prepare the revised paper. We merely note that the Robetta server is widely thought to generate reliable models.

We understand that when the original simulations were done AF2 was not available, still it will be good to generate now the model with AF2 and compare it with the one obtained with Robetta. This will be useful to validate and justify the used model. This would also be useful to justify this sentence from the main text: "Equally, the large fluctuations of the C-terminal tail in the replicated MD trajectories most likely are a consequence of these residues being mispositioned in the initial model ($t = 0$ ns)"

Q7: The authors performed MD simulations on both the monomeric and dimeric forms of the enzymes. However, what is the oligomerization state of the enzyme in solution? This could be determined experimentally, which will help then deciding which model is best to analyze and include in the main text.

A7: It is common practice to use a single monomer of a multimeric system in order to ensure that the simulations are completed in a reasonable amount of time and to facilitate analysis of the results. The oligomerization state of the enzyme in solution is irrelevant in our opinion because we are comparing the MD simulations to data from the EM-derived dimer structure.

Being experts in MD simulations, we can state that what it is a common practice is to run the MD simulations using the oligomerization state of the enzyme that adopts in solution. In fact, the MD simulations are performed in explicit water solvent, so to rationalize and get insights about functional states of the enzyme MD simulations should be performed using the set of conditions that most closely matches the experimental conditions. This also applies to oligomerization states, although we agree that the analysis effort and computational cost is higher. One of the strong points of MD simulations is that it is complementary and provides additional insights to what is observed in X-ray and EM-derived structures.

Q8: In Figure S18, a comparison of backbone RMSF values computed from 200 ns MD trajectories of the full-length apo-ASNS monomer and the apo-ASNS dimer is displayed. Although the authors mention that no significant differences were observed between monomer and dimer, the RMSF displayed in SI does show significant differences: the dimer appears to be much more flexible than the monomer, especially in the 480-540 region for monomer B.

A8: The differences between the dimer and the monomer are not significant (up to 1 Å is not really a difference). The most noticeable difference is indeed in the 480-540 region which is known to be highly flexible and anyway has no impact on the behavior of Arg-142.

In Figure S19 (now), differences of up to 2 Å can be observed in the 40-60 region (monomer A), and differences higher than 3 Å are observed in the 490-510 region (monomer B). The point that the flexibility of the 480-540 region has no impact on the Arg-142 behaviour is not demonstrated.

Q9: In Figure 5, large differences between the cryo-EM structures and MD data are also observed in the region around 475-500, and 200-230. Why is that? Where are the regions located and how important are for the catalysis? The same is observed when comparing apo and ternary complex MD simulations, thus indicating that these regions play a key role for binding and catalysis. The authors mention that the large deviations correspond to regions not solved in the X-ray structure, so again we wonder about the reliability of the generated model.

A9: We thank the reviewer for pointing out this problem in Figure 5. These two regions are known to be flexible, and indeed, the structural models derived from either X-ray crystallography or cryoEM show low resolution at these sections of the enzyme. As a result, the fluctuation differences between the 3DVA and MD-derived arise from the lower quality of the EM-derived density for residues adjacent to residues that are missing in the EM structure. We have added new text and modified the figure legend to clarify what is going on.

We understand that these regions are not solved or have a low resolution, but we are still wondering about the role of these regions in binding and catalysis.

Q13: The comparison of the tunnels between WT and variants is also not conclusive. In Figure 6b, the tunnel for the R142I variant is displayed, which shows that at the 20 and 140 ns snapshots there is actually no tunnel. They claim that the conformation of Ile is not regulating tunnel formation, so clearly the formation of the tunnel is affected by additional residues not considered in the manuscript. The analysis of the bottleneck of the tunnel and which residues are located close to this narrowest region might be helpful to provide additional insights.

A13: We have modified the text to clarify our interpretation. The key point that we were trying to convey was that the presence of the intermediate results in large structural changes in the tunnel even though the Ile-142 side chain adopts a conformation that would otherwise allow ammonia to access the tunnel. Given the reservations of the reviewer about the validity of these computational models, however, we consider that detailing the roles of multiple residues in determining the structural stability of the ammonia tunnel lies outside the scope of the paper. Moreover, omitting this discussion does not impact our key observation that EM can be used to observe the Arg-142 side chain in multiple conformations.

We disagree. The paper puts a lot of emphasis on the role of Arg-142 sidechain conformation in tunnel formation, and in fact, the authors have already performed CAVER calculations, so that the information on the residues defining the bottleneck of the computed tunnels is already included in the performed calculations. This information will provide very relevant information and will confirm their hypothesis based on a set of distances and visual inspection of the structures and MD data.

Reviewer #6

(Remarks to the Author)

Version 3:

Reviewer comments:

Reviewer #5

(Remarks to the Author)

In this new revised version of the manuscript, the authors have addressed our original concerns (but not all of them).

“We have included a comparison of our Robetta structure with the structure predicted by the AF3 in Supplementary Figure S17.”

The comparison of the Robetta and AF3 model included in S17 shows large differences at the C-terminal region, but also at other loops (201-220, 465-475) contained in the middle and end of the sequence. Unfortunately, none of these regions are solved in the X-ray or cryo-EM structure, which indicates that these are highly dynamical regions (in the RMSF plots these regions display also some flexibility). However, the large deviation of both structures again questions the validity of the Robetta model, as AF2 has shown to predict highly accurate folded structures especially if similar structures were included in the training set.

“There is every reason to believe that the biologically active form of human ASNS is a dimer as seen in the cryo-EM structure (as well as crystal structure) and we believe that we have provided evidence to support our choice of using the monomer in our MD simulations.”

If there is every reason to believe that the biologically active form is a dimer, there is no need to run the MD simulations with the monomeric form of the enzyme.

“There is no reason to believe that this region (480-540) will have any impact on Arg-142 side chain behavior, which seems to be controlled by local interactions involving the salt bridge networks already shown and discussed in the manuscript.”

As the authors know, conformational changes/mutations/effector binding/etc far away from the active site can have a large influence on enzyme function. It might very well be that this region has no effect on Arg conformation, but considering the allosteric nature displayed by many enzymes, there might be some reasons to believe otherwise. In any case, this region is very close to the 465-475 region that Robetta/AF3 predicted different conformations, again questioning the validity of the used model.

“While we can readily appreciate the reviewer’s assertion that detailing the roles of multiple residues by MD stimulations will provide additional insights into tunnel structure and dynamics, such an analysis lies outside the scope of the present study and is best addressed in a separate paper that will appear in a more specialized journal.”

As mentioned in our previous report, the authors have already computed the tunnels with CAVER, so the information of the residues defining the bottleneck of the computed tunnels is already included in their already performed calculations. Therefore, no additional calculations need to be performed, but rather check the bottleneck information in the output.

Reviewer #6

(Remarks to the Author)

Reviewer 1:

Summary: In this study, researchers used 3D variability analysis (3DVA) combined with atomistic molecular dynamics (MD) simulations to investigate the dynamic motions of human asparagine synthetase (ASNS). By solving the structure of ASNS and performing 3DVA, they suggest that a single side chain's dynamic motion (Arg142) regulates the interconversion between open and closed forms of an intramolecular tunnel. The opening of this tunnel allows for the translocation of ammonia, which is necessary for ASNS's catalytic function. MD followed up on this initial finding to determine exactly how.

The study highlights the power of cryo-EM in identifying localized conformational changes and demonstrates how conformational dynamics can regulate the function of metabolic enzymes with multiple active sites. However, the lack of experimental electron density shown in the figures (or available publicly) makes it difficult to assess the claims in this study. Additional forward tests of the importance of this blockage via mutagenesis may also uncover why it has to be regulated at all. If this is out of the scope of the current paper, it should be hypothesized and speculated upon in the discussion.

Major Points:

1. In your figures, Please show electron density and all individual atomic positions. This includes Fig. 1d, 2a, and all of Figure 3.

We have revised both Figure 2 and Figure 3 to show the EM density associated with these residues in the map. As to including EM density for the residues in Figure 1d, we found that this makes it difficult for the reader to appreciate the molecular interactions at the dimer interface. We have therefore created a new supplementary figure (Figure S8) that shows the EM density observed for these residues in the map.

2. Please show the PCA of the 3DVA. Clarify whether this was done on the entire structure or the tunneling residues. If done on the entire protein, please comment and show if other changes were seen elsewhere.

3D variable analysis (3DVA) was carried out using the entire enzyme in dimer form because the procedure is based on the variance of coordinates compared to a mask of the protein. We now state this point in the revised text. We have also added a supplementary figure (Figure S14) showing the five PCs for the apo-ASNS dimer as a function of latent coordinate vs. particle distribution as well as 3D representation of particle distributions. The same information has also been added for the R142I ASNS variant (Figure S27). If the reviewers would like a more detailed explanation of the method, then we refer them to the following paper:

Punjani, A. and Fleet, D. J. 3D Variability Analysis: Resolving continuous flexibility and discrete heterogeneity from single particle cryo-EM images. *J. Struct. Biol.* **213**, 107701 (2021).
<https://doi.org/10.1016/j.jsb.2021.107702>

Briefly, 3DVA generates 20 frames (20 different EM maps) for each principal component. Movements undergone by the main chain atoms in these 20 maps represent "motions" captured in a given principal component. These overall motions can be seen in the movies (M1-M5 for WT ASNS, and M6-M10 for R142I variant) provided as supplementary material, which also show that these movements take place across the ASNS structure. Given that the RMSD from superimposing the first and last frames is less than 1 Å, however, the overall movement appears

to be subtle. As our interest has been focused on the motion of residues near to, and within, the ammonia tunnel, particularly Arg-142, we have not yet investigated other motions and their potential roles in ASNS function. Such work will require the same level of scrutiny and analysis to that reported here for Arg-142. As a result, evaluating and discussing any other changes in the enzyme is beyond the scope of the work described in this manuscript.

3. In the RMSD analysis, please clarify what EM coordinates you are using. Are you comparing all structures from the 3DVA? Please also provide raw values as well as normalized values.

As described above, for each principal component, 3DVA generates 20 different EM maps (frames) capturing motions of ASNS in dimer form. Each frame (EM map) was then translated into a high-resolution structure (PDB format) using the “varref” software (*Biochim. Biophys. Acta Biomembranes* **1865**, 184133 (2023)). Coordinates were generated for 20 PDB files from each principal component to give 100 dimer structures that were processed using the CHARMM GUI () and CPPTRAJ to give a single “trajectory” file of the “Chain A” monomer. This file was then used to obtain RMS fluctuations for each residue in Chain A using standard methods implemented in CPPTRAJ. All raw and normalized RMSF values used to generate the figures in the paper are provided in an Excel datafile that is now provided as supplementary material.

4. Your results do not support the claim ‘Our results suggest that changes in the C-terminal active site are propagated over a distance of approximately 20 Å, leading to tunnel opening and ammonia translocation’. While the data here shows that the tunnel can move between an open and closed state in apo form as part of the native fluctuations (revealed by the PCA analysis). There is no information presented on how this information is propagated nor how the active site or binding interacts with this motion. Please change the wording of this or explain the mechanism.

We agree with the reviewer and have made textual changes that clarify this comment, which is no longer included in the abstract. These are indicated in the “marked up” version of the revised MSS by text in red font.

Minor Points:

1. Neither the PDB nor the map is publicly available, making it difficult to examine the structures and map independently. Please release them. Also, include information and metrics regarding map sharpening and map-to-model fit. Zenodo is a good option for the 100 structures from the PCA analysis.

We have deposited the EM-based coordinates for the apo-ASNS dimer and the R142I ASNS variant into the PDB and the Electron Microscopy Data Bank (accession numbers are provided in the data availability section of the revised manuscript). Following well-established, standard practice, these structures and other raw data will be released on publication of this paper. In order to give the reviewers full access to this information, however, we have placed the following files into a publicly accessible Dropbox folder:

- 1) The consensus EM maps (EMD-40764, EMD-44253), and the models for WT (PDB ID: 8SUE), and R142I variant (PDB ID: 9B6C)
- 2) 100 3DVA-derived EM maps and the corresponding PDBs for the WT and R142I variant. Note that 5 PC x 20 maps and PDBs = 100 total for WT and for the R142I variant. For each principal component, 20 3DVA-derived maps are packed into one zip file.

These datasets can be accessed using the following link:

<https://www.dropbox.com/scl/fo/z6r4xaau9m9cdo1p7rnoz/APqZL7kHNng1VAEPCcHBEG1c?rlkey=5uzhk0jyvmgloqdrhxayl6fy&dl=0>

2. In Figure 1d, please label the amino acids and chains. Please provide experimental density corresponding to the positions of these residues in this figure or a supplementary figure.

We have labelled the amino acids and chains in a revised Figure 1. As noted above, placing all the information (EM density, amino acids and chains) onto a single figure makes the figure unreadable. Thus, amino acids and chains are labelled in Fig. 1d, and the model with EM density is displayed in Supplementary Figure S8.

3. In Figure 2a, please provide a legend for what each color represents.

We have revised Figure 2, and the legend has been modified appropriately to address this reviewer query.

4. How did you determine 5 PCAs for the 3DVA analysis?

As implemented in cryoSPARC, the user specifies how many principal components are to be computed in the 3DVA, which are then sorted in order of importance. Obviously, 3 components are the minimum number as they represent 3 orthogonal coordinates. In our case, we thought that 5 components should be sufficient to see any relevant movements, assuming these existed. Given that we did observe functionally relevant movements, we believe that this choice is correct.

5. Please provide details on the normalized RMSF. How was this normalization done?

RMS fluctuations for each residue were computed for each of the MD trajectories and for the 100 structures produced in the 3DVA (these structures can be treated as another trajectory). Each set of RMSF values for each trajectory was then divided by the largest RMF value in that trajectory to give a normalized set that ranged from 0-1. All raw and normalized RMSF values used to generate the figures in the paper are provided in an Excel datafile that can be downloaded from the *Nature Communications* website.

6. In Figure 4, please provide a legend for all colors of amino acids and tunneling.

We have revised Figure 4, and the legend has been modified appropriately to address this reviewer query.

7. Please deposit the coordinate files for the 100 structures used in the 3DVA study and (ideally also two MD-derived trajectories) on Zenodo or similar repository.

As discussed above, we have placed the 200 structure files from the 3DVA analysis of both WT ASNS and the R412I variant, together with all of the MD trajectories discussed in the paper, into a Dropbox folder. These datasets can be accessed by the reviewers using the following link:

<https://www.dropbox.com/scl/fo/z6r4xaau9m9cdo1p7rnoz/APqZL7kHNng1VAEPCcHBEG1c?rlkey=5uzhk0jyvmgloqdrhxayl6fy&dl=0>

Reviewer 2:

The manuscript reports cryo-EM model and associated 3DVA and MD simulation analysis of human asparagine synthetase. Beyond some verifications between MD and cryo-EM, a key result is the observation of closed conformation in the 3DVA analysis even when the ligand is absent. My comments are limited to the MD simulation part of the work.

First, the idea of a protein conformational pre-equilibrium prior to substrate binding is already very well established. So it does not strike as a surprise unless the functional relevance for the synthetase is sketched out.

We have modified the manuscript to clarify our comments about conformational selection.

Second, the MD simulation is weak in that only one part of the transition is visible, which statistically means very little. Unless in simulations the entire 'open to close back to open' conformations is observed multiple times, the mentioned dynamic pathway of closing have minimal relevance. Extended sampling tools like CryoFold (Matter 2021 by Shekhar et al) are meant for this purpose. The movements from 3DVA similarly also has to be verified by tools like manifoldEM or Polaris to have any interpretation of the dynamics in terms of the energy landscape.

A microsecond-scale trajectory in which the tunnel interconverts between the open and closed forms has been performed and is now discussed in the revised manuscript. We believe that addressing the manifold question and calculating the energetic landscape lies outside the scope of the present paper. In unpublished comparisons of MD and 3DVA, we observe that directions and space exploration are similar but MD being less restrained goes further in the movements.

CryoFold (or now CryoFold 2.0) is a program which enables the construction of atomic models out of EM maps with "resolution" heterogeneity, i.e. where the density data is of varying sparsity at 3–5 Å resolution, or worse. We point out that the overall resolution of our EM map is 3.5 Å for WT and 3.3 Å for the R142I variant, and that the local resolution in the region of the ammonia tunnel is around 3.0 Å for WT and R142I variant. Thus, our EM map does not suffer resolution heterogeneity. The "varref" software (stands for "variability refinement") has been integrated into the internationally recognized, and widely used, program Phenix. Varref or phenix.varref allows refinement of structures with variability from a consensus atomic model. The ensemble of maps represents the sample's "conformational" heterogeneity that is captured during the experiment. The bundle of structures generated by varref make it easy to spot variable parts of the structure, as well as regions that are not moving. We therefore believe that varref is particularly well suited for analyzing the map series derived from 3DVA (cryoSPARC), or multibody refinement (Relion). For these reasons, we do not see any need to use CryoFold in our work, especially because our EM map does not suffer from resolution heterogeneity.

Third, so much is already known from crystallography that the cryo-EM piece looks incremental at best. This can be a presentational issue but dampens my excitement about the findings.

As we point out in the paper, there were multiple concerns about the biological relevance of the X-ray crystal structure, including whether the observed ASNS dimer was artefactual. We have made some textual changes, however, to clarify this point further.

Finally it was already shown by Schulten and co-workers that the fluctuations in MD and those from the maps when comparable can offer a reasonable map-modeling criteria. The work should be acknowledged and indeed offers support to the article.

We were unaware of the work by Schulten and Tama. We have therefore updated the main text to include relevant ideas and literature citations. We thank the reviewer for drawing our attention to this important omission.

Reviewer 3:

The authors use 3D variability analysis (3DVA) of an EM structure for human asparagine synthetase (ASNS) in combination with atomistic molecular dynamics (MD) simulations to detail how dynamic motions of a single side chain mediates interconversion of the open and closed forms of a catalytically relevant intramolecular tunnel. This interconversion between open and closed states allows ASNS regulating catalytic function. The findings demonstrate the utility of cryo-EM in providing information on protein motions, even at the level of a single residue in an enzyme possessing multiple active sites. This is excellent work which is presented in a clear and distilled manner and in my opinion meets the high requirements of Nat. Commun. However, the authors should carefully consider the suggestions below. Especially the point regarding performing additional replica is crucial.

We thank the reviewer for these very positive comments. We have indeed performed the replica calculations and related changes to the manuscript are discussed in more detail below.

1. Abstract:

- a. Sentence logic unclear: "How dynamical motions in enzymes might be linked to catalytic function is of significant general interest, although almost all relevant experimental data, to date, has been obtained for enzymes with a single active site."
- b. "dynamic motions": Isn't all motion dynamic? Remove redundancy.

We have substantially revised the abstract to address these concerns. Revisions are indicated by text in red font.

2. Introduction:

- a. "Moreover, experimental evidence has been published that supports the existence of thermal networks in enzymes, which mediate energy transfer between the active site and external solvent.". I don't think this statement is supported by actual evidence in the cited articles. This is a proposal, which is lacking any real hard evidence. Commonly, we assume equipartition of kinetic energy in classical system, like heavy atoms in proteins. I suggest modifying the sentence to reflect better what has been actually proven scientifically.

Yes, the reviewer brings up a good point. In addition to other additions/corrections in the Introduction that are detailed below, we have modified the offending sentence to address these concerns.

3. Results:

- a. The residues involved in interactions stabilizing the closed and open forms must be included and marked in Fig. 3, 4.

We have included a new supplementary Figure S29 that shows these interactions.

b. MD simulations: The events seen in the MD simulations could be happening by chance, as this is a deterministic method. The authors must perform an additional 2-4 MD simulations with different initial conditions (e.g., velocities) of both systems simulated to obtain statistically significant data, to reach 3-5 replicas that is the accepted number for vanilla MD simulations. See f. ex. Knapp, B., L. Ospina, and C.M. Deane, Avoiding False Positive Conclusions in Molecular Simulation: The Importance of Replicas. *Journal of Chemical Theory and Computation*, 2018. 14(12): p. 6127-6138.

We agree and have now performed replicate MD simulations for WT ASNS and two ASNS variants. The results, which are now discussed in the main text and the SI, support the conclusions of the original submission.

c. It would also be good to extend the length of the simulations to verify that the conformational transitions are reversible on the time scale of the simulations.

We agree and have now performed extended, microsecond MD simulations for the two WT systems. The results, which are now included in the main text and the SI, show that Arg-142 undergoes multiple changes between the orientations that open and close the ammonia tunnel over the microsecond timescale of the simulation.

d. Why was diffusion of NH₃ not modeled? This is fairly straightforward.

We believe that such calculations lie outside the scope of this study. The focus of the present paper is really to show the power of 3DVA methods in identifying how functionally important side chains move within (relatively small) enzymes.

4. Methods:

a. "The hydrogen bonding networks in each of the models were then optimized, and the protonation state of ionizable residues at pH 7.4 determined by calculation." The type of calculation used must be specified.

Yes, we agree. The protonation state was assigned using PropK_a, and this is now clearly explained in the methods section. In addition, we now include details in Supplementary Table 2 about the protonation state of ionizable residues in all of the models used in the MD simulations.

Reviewer 4:

The manuscript by Richards and colleagues combines cryo EM studies with MD simulations to obtain insights into dynamic processes that regulate the activity of the Human Asparagine Synthetase enzyme (ASNS).

As the authors write, it is highly important to understand how motions in enzymes are correlated with function. Modern methods in structural biology have provided a very large number of static images of bio-molecular complexes. These static images are important, but fail to provide information on how structural changes in these complexes regulate function. In that light, the current study is an important example that links the motion of a single side-chain with ammonia transfer from one active site in ASNS to the other one.

Despite this importance, I have a number of major remarks to the manuscript. In many cases I am not sure if the conclusions are sufficiently covered by the (limited) data. To fully claim the

proposed model, additional cryo EM structures, e.g in the presence of substrates, would be helpful/needed.

We thank the reviewer for these comments, and for their very thorough review of our initial submission. Addressing their criticisms has indeed improved the quality of the paper (see below). Despite our best efforts, we have so far failed to obtain any cryo-EM structures in which ligands can be observed in the enzyme, with the exception of the DON-modified human ASNS.

1. The authors mention at multiple locations that NMR spectroscopy is limited. I do agree with that to some degree, but recent advances have made NMR amenable to study enzymes with multiple active sites (<https://doi.org/10.1038/s41467-021-22968-6>) or of very large complexes (<https://doi.org/10.1126/science.1233066>, <https://doi.org/10.1038/s41589-022-01111-6> or <https://doi.org/10.1038/nature12581>) to mention a few random examples.

Thanks for these valuable comments and citations. We have sharpened up the relevant text to point to these advances in NMR methods and to include these citations.

2. Did the authors try to do a template-based particle picking with the bacterial homolog (AS-B)? This is important in case the authors would like to claim that other dimers are not present.

We have now included a discussion of our efforts to perform template-based particle picking with the AS-B dimer in the main text and added a supplementary Figure S4. As we now point out in the paper, this merely resulted in the selection of a large number of junk particles. We therefore conclude that other forms of the human ASNS dimer are not present.

3. Is it true that only 13.2% of the particles are the dimeric enzyme. If yes, does that not imply that the enzyme exist in a monomer \leftrightarrow dimer equilibrium. Is that also observed in solution (is there SAXS or SEC data)? Is the dimerization relevant for catalysis? This should be clearly stated, as currently I am not sure what the consequence of the (kind of) dimerization is. Does the “monomer” have the same R 142 behavior regarding the tunnel?

The reviewer brings up an important point. There is, however, no published evidence to suggest that enzyme activity depends on the dimerization of ASNS monomers. We have modified the text to make this point more clearly, as suggested by the reviewer. We have also performed MD simulations of the WT ASNS models to examine whether dimerization affects the motions of Arg-142. Our results, which are now included in the Supplementary Material indicate that motions in the ASNS monomer are unchanged compared to those of monomers in the ASNS dimer (Figure S18).

3. “ it is likely that these regions are disordered rather than being absent due to proteolysis.” I agree that regions that are not well visible are likely not well ordered. Is there any indication of proteolysis (e.g. from an SDS gel).

When we determined the structure of ASNS by X-ray crystallography (Zhu et al., 2018), it took several weeks for ASNS to crystallize. Thus, it was possible that two loop segments (residues 201-220 and residues 465-475) might have been proteolyzed during the crystallization process. However, this is not case for our cryo-EM studies for two reasons: first, at least as judged by SDS-PAGE analysis of recombinant WT ASNS and the two ASNS variants used in this study, we see no indication of proteolysis (Figure S10). Second, in preparing cryo-EM specimens, both enzyme samples were thawed, spotted onto EM grids, and vitrified by immediately introducing each grid into liquid ethane. As a result, there was insufficient time for proteolysis to take place.

4. “especially because this residue, Arg-142, is conserved in mammalian asparagine synthetases” This sounds more impressive than it is, as almost all residues are conserved in mammals according to figure S8. As the authors claim an important role for R142 in the regulation of the access to the ammonia tunnel the question arises how this important aspect takes place in the bacterial enzymes (that don’t have the Arg in that position). How does the Arg (mammalian) to Ile (bacterial) mutation in the human enzyme effect activity/ regulation?

The reviewer makes a good point, although a complete functional characterization of tunnel related ASNS variants lies outside the scope of this paper. We now, however, report preliminary work aimed at investigating the role of Arg-142. Thus, we have expressed and purified the R142I and R142A ASNS variants and characterized their kinetic properties. In addition, a cryo-EM structure has now been determined for the R142I ASNS variant. These experiments highlight the differential effects of changing Arg-142 when L-glutamine and ammonia are used as nitrogen sources and are now described in the text of the revised manuscript.

Our findings, however, can be briefly summarized here: The R142A and R142I ASNS variants exhibit similar steady-state kinetic behavior. In both cases, the ability of the enzyme to form the beta-aspartyl-AMP intermediate is impaired when L-glutamine is the nitrogen source but is effective unaffected when ammonia is present in the assay. These findings therefore support the functional importance of Arg-142 for catalytic turnover in human ASNS. On the other hand, MD simulations and the cryo-EM structure of the R142I ASNS variant suggest that the tunnel adopts an open conformation in both variants. At first glance, the idea that unimpeded ammonia translocation between the active sites might be expected to enhance activity but this is likely not the slowest step in the mechanism. Two reasons are suggested by the R142I structure and the MD simulations. First, the absence of Arg-142 will break the network of salt bridges thereby mis-positioning residues in either the synthetase or glutaminase sites, which may impair glutaminase activity or the ability of the variant to form beta-aspartyl-AMP.

In essence, even though replacing Arg-142 by isoleucine “opens the gate”, it also leads to closed and/or abnormal tunnel structures, thereby resulting in decreased ammonia availability for reaction in the synthetase active site. Our data from cryo-EM structure, MD simulations and kinetic studies are all consistent with this hypothesis. Although we recognize that elucidating the details of how replacing Arg-142 impairs the enzyme is of great interest, we believe that such an investigation lies outside the scope of this paper.

5. There is some density shown in figure S5. I would appreciate a clear view of R142 in addition.

We have modified this figure to provide a clear view of R142, as requested. This can be found in supplementary Figure S7a in the revised version of the supporting information.

6. “In contrast, the intramolecular tunnel is continuous when glutamine and AMP are bound within the active sites of the C1A AS-B variant (i.e., it is in an “open” form).” The authors compare two different things here: the apo human structure (closed), with the E. coli in the presence of AMP and glutamine. It is thus not clear if the AMP binding caused the opening of the tunnel (as the authors suggest) or if the tunnel opening is due to the glutamine binding or the differences in amino acid sequence. One should really compare the tunnel opening of the human protein in the presence and absence of (one or two) ligands, preferably by cryo-EM.

In principle, we agree with the reviewer. Unfortunately, and notwithstanding considerable effort, we have so far failed to identify conditions that yield a cryo-EM structure of human ASNS bound

to substrates or non-covalent inhibitors. Indeed, that is, in part, why we turned to computational simulations.

6. The 3DVA analysis is impressive, but I would need to see densities that would allow for the unambiguous placement of the R142 side chain. At the resolution of the structures, I am not sure if it is possible to place R142 accurately. What percentage of the complex is open/ closed?

The results from 3DVA are summarized in Fig. 3. We have revised the original version of this figure to include the EM density for corresponding residues. For this revision, we note that 3DVA was conducted using a resolution filter of 4 Å, as suggested in the cryoSPARC manual. Indeed, a use of a resolution filter at 3 Å did not make the maps any better. As shown in our revised Fig. 3, the quality of EM densities is good enough to distinguish between the open and closed locations of the R142 side chain.

In the 5 PCs, two components correspond to “open”, two to “closed” and one to “partially open/closed”. Based on these observations, the percentage of the complex being open or closed is estimated to be 50%.

7. The 3DVA analysis suggests that the apo protein (that the authors study here) is partially in an “tunnel open” state. That makes little sense to me, as then ammonia could “escape”. In case the “plug” is only closed part of the time the complete regulation is non-functional.

As noted in the paper, ASNS is unusual for a type II glutamine-dependent amidotransferase in that an unusually high, intrinsic glutaminase activity is present even in the absence of L-aspartate and MgATP. Thus, ammonia can be released when L-glutamine binds to the enzyme in the absence of L-aspartate (for more details, see ref 57). In any event, not all of the conformational states observed in apo-ASNS by 3DVA may be functionally relevant.

8. Is the MD done on a monomeric form of the enzyme? That appears odd, as the authors clearly show a dimeric form in the cryo-EM.

We have performed an MD simulation of the ASNS dimer. As noted above, and in the supplementary material, this new trajectory shows that the motions of the ASNS monomer are equivalent to those of either monomer in the ASNS dimer. This is not too surprising given the relatively small region of contact between the monomers.

9. To me there are no significant differences in the tunnel opening between the apo and ternary complex. The MD simulation is too short to draw conclusions regarding the statistics of tunnel opening/ closing. In Fig. 4C the tunnel appears actually wider in the apo state. In this figure the two active sites as well as the start and end of the tunnel should be clearly indicated.

We now address this point using microsecond MD simulations, which confirm that numerous interconversions between the open and closed states of the intramolecular tunnel can take place.

10. The link between the substrate binding and tunnel opening is described in the last part of the results, but this is not clearly shown in a figure. How does this “look”. Also, the open state should be structurally compared to the open state in the bacterial enzyme.

A new figure in the supplementary material (Figure S19) now provides a visual comparison of the open form of the tunnel in the human ASNS/beta-aspartyl-AMP/MgPP_i ternary complex (after 200 ns of MD simulation) and that seen in the X-ray structure of AS-B, the bacterial homolog.

11. The importance of R142 in channel closing should be shown experimentally, e.g. by performing activity assays that assess if the glutaminase activity is larger when the tunnel is open (R to A mutation?).

As suggested by the reviewer, we have expressed and purified the R142A ASNS variant, and report its steady-state kinetic behavior in the revised paper (Table 1). In addition, we prepared the R142I ASNS variant reasoning that R142I should form the open conformation of the tunnel, as observed in the X-ray structure of the bacterial homolog, AS-B. Although these variants possess glutaminase activity, they have altered steady-state kinetic properties compared to WT human ASNS (Table 1). We have therefore revised the paper to include MD simulations identical to those reported for the WT enzyme and have determined the EM structure of R142I ASNS variant as well. These new findings are now included in the text of the revised manuscript and highlight the differential effects of changing Arg-142 when L-glutamine and ammonia are used as nitrogen sources. More details are provided above.

12. “The role of Arg-142 in controlling conformational plasticity in the intramolecular tunnel of apo-ASNS contrasts sharply with observations on other Class II amidotransferases in which substrate binding is essential for tunnel formation and activation of glutaminase activity”. Please explain this in detail, as also in the present case substrate binding seems to be important in opening the channel. And how do the authors know that binding of additional substrates has no influence on the channel, there is no EM or MD data regarding this.

The reviewer may be over-interpreting our statements, perhaps because we did not specifically refer to type II glutamine-dependent amidotransferases. We have therefore made textual revisions to explain this point more clearly.

13. Are any domain motions or re-orientations observed in the MD, as has been observed for related complexes?

We observe no significant domain motions or re-orientations in the MD simulations. In having the tunnel present in both apo- and liganded forms, ASNS seems to be unique in the family Class II amidotransferases.

14. “Our results suggest that changes in the C-terminal active site are propagated over a distance of approximately 20 Å,”. I don't see this. The authors show some very local changes only, e.g. in Fig. 4C.

Yes, as discussed in our response to reviewer 1, this statement really refers to the idea that a local motion impacts the function of a tunnel of approximately 20 Å in length. We have revised the text to clarify this point.

Reviewer 1:

Dropbox is not a good solution for storing the data as they have done. Suggest using zenodo or dryad.

Our use of Dropbox was a temporary solution used to satisfy the requests of this reviewer for access to the density maps without violating confidentiality. As they suggest, we will deposit all 3DVA-related data, and simulation trajectories and related files into Zenodo on acceptance of the paper for publication in Nature Communications. Movie files will be made available via the Nature website.

Reviewer 2:

The new analysis definitely strengthens the article and suggests that the movements seen in 3DVA might have some physical relevance as seen in MD. But unless a free energy analysis of the MD simulations is performed to indeed show that the energy changes (including barriers) associated with the conformational are thermally accessible, the data will remain weak.

Although the data in the microsecond simulations suggest that the Arg-142 side chain moves between the open and closed conformations identified in the EM map, we have performed additional metadynamics simulations to obtain a picture of the free energy surface for changing the conformation of the Arg-142 side chain in both apo-ASNS and the associated ternary complex. Not only do these results show the relative energetics of the open and closed conformations but they provide a quantitative estimate of the barrier for interconversion that supports the idea that this process is thermally accessible at 298K. In addition, they support the existence of a third conformer corresponding to the partially open form (Fig. 3). Relevant citations, Supplementary Figures (S15 and S21) and methods for these calculations are now included in this revision of the manuscript. We thank the reviewer for their suggestion.

Reviewer 3:

This is excellent work which is presented in a clear and distilled manner and in my opinion meets the high requirements of Nat. Commun.

Although this reviewer did not comment on the revisions made to address their criticisms of our original submission, we draw attention to their very positive assessment of the study (considerably improved and extended in the revised paper), which they considered meets the high requirements of the journal. This assessment clearly conflicts with the opinion of reviewer 5 regarding innovation and general interest (see below). We believe that the assessment of reviewer 3 would be even more positive about this version of the manuscript, which has been revised to address all of the points raised in their original review.

Reviewer 4:

The authors have made significant efforts to improve the manuscript and have now presented a very convincing version where my initial criticism has been fully resolved. I especially appreciate the activity assays that have been included and that confirm the computational work. Note that I really appreciate the computational work, but that I am not an MD expert and can thus not 100% ensure that all computational details are fully correct. In case the other reviewers that might be experts in MD agree I support publication of the manuscript without further edits

We thank the reviewer for their positive assessment of the revisions made to the original submission. Our response to criticisms raised against the computational details are presented elsewhere.

Reviewer 5:

In this paper, Coricello and coworkers report cryo-EM and MD simulations of Human Asparagine Synthetase and identify Arg-142 as key residue for regulating the ammonia tunnel. Considering that some X-ray data was already available, and that the new cryo-EM and MD data are not providing enough evidence of the reported claims, the paper lacks the novelty required for publication in Nat Commun. Additionally, and more importantly, the conclusions withdrawn are not well supported by the MD data, and the analysis of the simulations is far from optimal.

Despite the fact their overall assessment sharply contrasts with that of all other reviewers, we have benefitted from this very comprehensive review of our work. There is no doubt that the revisions made to address these concerns have improved the manuscript. On the other hand, we would like to point out that the novelty of our study lies in the fact that 3DVA can yield information on conformational changes at the level of a single side chain and is therefore useful in generating hypotheses about the functional importance of conserved residues. The X-ray structure of human ASNS provides no clear understanding of how the Arg-142 side chain might act to control ammonia access to the tunnel.

It is really surprising that a paper focused on the analysis of conformational dynamics by means of cryo-EM and MD contains an introduction with only a few citations to computational papers covering the analysis of the conformational landscapes.

Almost all of these papers focus on allosteric networks, which generally impact substrate binding (K_m) rather than catalysis (k_{cat}). We agree, however, that these papers should be included and have revised the introductory text as requested by the reviewer. As a result, we have also added new citations (8, 13, 14, and 31-35).

The authors analyze the effect of Arg-142 conformation as defined as open/closed (or in some cases partially open), on both the X-ray data and some snapshots along the MD simulations performed. How is this open/closed defined? Based on a distance, angle, dihedral of the Arg-142 sidechain? How do they know that Arg-142 conformation is directly related to tunnel formation? In the included Figures (4, 6) Arg-142 is not displayed, so such correlation cannot be extracted. It is therefore not clear how the authors know that Arg “open/closed” regulates tunnel formation.

The definition of open and closed is based on whether the side chain blocks access to the tunnel (closed) or not (open), as seen in the EM structure. The key idea here is that Arg-142 gates the access of ammonia to the tunnel, whether the latter is in a continuous or discontinuous form. We have attempted to make this point clearer by using more precise language in the revised manuscript. We now also indicate the location of either the Arg-142 or Ile-142 side chain using red dots in Figures 4 and 6. A quantitative description of the two conformations is also now presented in Supplementary Figure S15, which forms the basis of our choice for the two collective variables used in the new metadynamics simulations (Supplementary Figure S21).

It is stated in the text: “Although no change is seen in two PCs (components 0 and 2) (Figs. 3b and 3d), the Arg-142 side chain is re-positioned in two other PCs to yield an “open” form of the

tunnel.” Which are the main coordinates contributing to the different PCs? Only the tunnel residues are displayed, but many other changes might be observed in the protein structure as shown in the attached movies. It is also stated that the PCA is done considering the coordinates of the entire enzyme in the dimeric form, so the contribution of the Arg-142 conformational change might be rather low in all described PCs.

In 3DVA analysis, per-particle variance compared to the protein mask is computed. The resulting latent coordinate space is further de-convoluted by Principal Component (PC) Analysis and analyzed linearly. Protein movements are thus "sorted" by order of importance of the variance contribution in the particle stack analyzed. In consequence, the whole structure can be considered as moving in all of the individual PCs, but differently. As indicated by the reviewer, these movements are visualized in the movies (M1-M5 for WT ASNS and M6-M10 for the R142I variant).

Given how the protein moves in each PC, the backbone atoms throughout the protein move differently, which results in the observed conformational “flip” of the Arg-142 sidechain. The fact that the latent space is very similar in each PC (Figures S14 and S27) confirms this analysis. Moreover, because we are comparing comparable objects in latent space, the particle stacks are also similar in terms of overall protein movement. Therefore, local changes that are observed do indeed reflect differences in the protein's conformational preferences. 3DVA analysis does not therefore look at individual side chain movements. It was therefore our choice to focus on residues involved in tunnel function in this study, **particularly Arg-142**.

The sidechain movement of Arg-142 observed in the component 0 is what we term “closed”, component 1: “open”, component 2: “closed”; component 3: “open”; component 4: “partially open” (as we show in Figure 3). What is important here, however, is that these observations led to the insight that Arg142 can act as the gate for the tunnel. This paper is therefore about how we tested this hypothesis using MD simulations, site-specific mutagenesis, cryo-EM and kinetic assays. We have, however, modified the text throughout this revision of the manuscript to emphasize this point.

The authors mention at the end of the results section: “This level of agreement between experiment and theory strongly supports the EM-derived mechanism in which movement of the Arg-142 side chain results in tunnel opening and ammonia translocation.” In our opinion, such statement cannot be made as the correlation between Arg-142 conformation and tunnel formation has not been demonstrated.

We have removed the offending statement as part of our efforts to clarify our interpretation of the experimental and computational data.

“Using the Robetta server, we constructed a model of full-length human apo-ASNS”. It is not clear to us why the more recent and accurate AF2 multimer tool was not used to reconstruct the full model. How reliable is this Robetta structure?

AF2 was not available when we performed the original simulations and we sought to be consistent in subsequent calculations used to prepare the revised paper. We merely note that the Robetta server is widely thought to generate reliable models.

“The resulting model, which was independent of the cryo-EM structure”. It is not clear what the authors mean with this statement. How reliable is the model as compared to cryo-EM structures?

Again, we have modified the text to remove this misunderstanding. We did not use information from EM and so the observed motions in the MD simulations are unbiased by the experimental data to which they are compared.

The authors performed MD simulations on both the monomeric and dimeric forms of the enzymes. However, what is the oligomerization state of the enzyme in solution? This could be determined experimentally, which will help then deciding which model is best to analyze and include in the main text.

It is common practice to use a single monomer of a multimeric system in order to ensure that the simulations are completed in a reasonable amount of time and to facilitate analysis of the results. The oligomerization state of the enzyme in solution is irrelevant in our opinion because we are comparing the MD simulations to data from the EM-derived dimer structure.

In Figure S18, a comparison of backbone RMSF values computed from 200 ns MD trajectories of the full-length apo-ASNS monomer and the apo-ASNS dimer is displayed. Although the authors mention that no significant differences were observed between monomer and dimer, the RMSF displayed in SI does show significant differences: the dimer appears to be much more flexible than the monomer, especially in the 480-540 region for monomer B.

The differences between the dimer and the monomer *are not* significant (up to 1 Å is not really a difference). The most noticeable difference is indeed in the 480-540 region which is known to be highly flexible and anyway has no impact on the behavior of Arg-142.

In Figure 5, large differences between the cryo-EM structures and MD data are also observed in the region around 475-500, and 200-230. Why is that? Where are the regions located and how important are for the catalysis? The same is observed when comparing apo and ternary complex MD simulations, thus indicating that these regions play a key role for binding and catalysis. The authors mention that the large deviations correspond to regions not solved in the X-ray structure, so again we wonder about the reliability of the generated model.

We thank the reviewer for pointing out this problem in Figure 5. These two regions are known to be flexible, and indeed, the structural models derived from either X-ray crystallography or cryo-EM show low resolution at these sections of the enzyme. As a result, the fluctuation differences between the 3DVA and MD-derived arise from the lower quality of the EM-derived density for residues adjacent to residues that are missing in the EM structure. We have added new text and modified the figure legend to clarify what is going on.

The accuracy of the MD simulations of the ternary complex is also not clear. As shown in Figure S29, the PPi and beta-aspartyl-AMP are displaced along the 200 ns MD simulation. In fact, one might expect the phosphate group to be strongly bound to the active site, but instead it is substantially displaced. An additional point is that focusing the analysis on the first and last frame of the MD data is not optimal, how stable are the t=0 and t=200 ns snapshots? Why these should be taken as reference?

The point of Figure S29 is merely to illustrate how the altered network of salt bridges changes when the intermediate is present in the synthetase site of ASNS. This criticism is the result of the orientation of residues shown in the figure. Examination of the two structures shows that MgPPi stays bound to the P-loop even though the intermediate changes conformation. We have modified the text to address this point.

Figure S16 shows the plot of the distances between Arg-142/Glu-414 (a), and Arg-142/Asp-405. In none of the replicas, a short distance between Arg-142/Glu-414 is shown in the displayed snapshot a distance of 2.80 Å is highlighted, however, all replicas present a very long distance of ca. 8 Å in all cases. Similarly, the distance between Arg-142/Asp-405 is maintained around 3 Å in only two of the four replicas. This is indicating that in two replicas of the MD, Arg-142 is neither interacting with Glu-414 nor Asp-405. Which interactions is therefore Arg-142 making?

We thank the reviewer for raising this point. There is indeed an error in the legend for Fig. S16 and related figures; the attribution of the distances is incorrect. We have therefore fixed this problem throughout the SI. As pointed out by many researchers, particularly Warshel and Kamerlin, electrostatic interactions are long range and thus the Arg-142 side chain can still interact with the negatively charged carboxylates at 8 Å distances. We have also added a new Supplementary Figure (S15) to highlight how the Arg-142 interacts with Asp-405 and Glu-414 in the two conformations. Indeed, these distances are now used as the collective variables for newly included metadynamics simulations.

“Forming the Arg-142/Asp-405 salt bridge requires breaking a network of intramolecular interactions, involving Arg-142, Glu-414, Arg-415 and Asn-74, that is present in the initial ($t = 0$ ns) structure. Given that the Asn-75 side chain forms the oxyanion hole required for glutamine hydrolysis, breaking this network provides a potential molecular mechanism by which tunnel opening might be coordinated with ammonia release in the glutaminase active site.” None of these additional interactions are displayed in any of the figures in the main text and SI, which only show the Arg-142/Asp-405 salt bridge. It is therefore not clear how the authors conclude such statement.

We agree, and so an additional Supplementary Figure (S15) has been prepared to illustrate the involvement of the Asn-74 side chain in the network that stabilizes the open conformation of Arg-142.

The comparison of the tunnels between WT and variants is also not conclusive. In Figure 6b, the tunnel for the R142I variant is displayed, which shows that at the 20 and 140 ns snapshots there is actually no tunnel. They claim that the conformation of Ile is not regulating tunnel formation, so clearly the formation of the tunnel is affected by additional residues not considered in the manuscript. The analysis of the bottleneck of the tunnel and which residues are located close to this narrowest region might be helpful to provide additional insights.

We have modified the text to clarify our interpretation. The key point that we were trying to convey was that the presence of the intermediate results in large structural changes in the tunnel even though the Ile-142 side chain adopts a conformation that would otherwise allow ammonia to access the tunnel. Given the reservations of the reviewer about the validity of these computational models, however, we consider that detailing the roles of multiple residues in determining the structural stability of the ammonia tunnel lies outside the scope of the paper. Moreover, omitting this discussion does not impact our key observation that EM can be used to observe the Arg-142 side chain in multiple conformations.

In Figure S19, a comparison of the open tunnel seen in the X-ray structure of AS-B (wheat) and the open tunnel of human apo-ASNS predicted by the MD simulations (grey) is displayed. Although the authors claim that the tunnels are similar, this is hard to tell with what is displayed in Figure S19.

This Supplementary Figure and associated text have been removed from the manuscript.

“One might expect no impact on catalytic activity under our assay conditions because ammonia translocation is unlikely to be the rate-limiting step in the catalytic cycle.” Which is the rate-limiting step along the cycle? How do the authors know that there is no change in the rate-limiting step after mutation, according to their simulations substantial differences in tunnel formation are observed between WT and variant.

The catalytic efficiency and k_{cat} in the ammonia-dependent assays are essentially unchanged for the three enzymes, which indicates that very little has happened to the energetic barriers for the chemical steps throughout catalytic turnover. The difference in behavior of the variants relative to WT ASNS in the glutamine dependent activity is therefore likely associated with the ‘availability’ of ammonia in the C-terminal active sites, as suggested by the MD simulations. This criticism, however, is not germane to the main focus of the paper, i.e. that using 3DVA on EM maps for enzymes can yield insights into functionally important conformational changes, which can then be investigated by MD simulations and experimental assays.

“Whatever the molecular basis for the altered steady-state kinetic parameters of the two ASNS variants, however, these kinetic studies support the hypothesis that Arg-142 is important for tunnel function, L-glutamine recognition, and the structural integrity of the tunnel given that free ammonia likely accesses the synthetase site directly rather than passing through the intramolecular tunnel.” What is the molecular basis of the effect of the mutation? It is not clear in their interpretation of the kinetic data.

Our molecular hypothesis is that the tunnel is deformed during catalytic turnover when R142 is absent, and hence the ammonia translocation is impaired, as suggested by the MD simulations of the R142I/intermediate complex. Regardless, the kinetic data clearly show that the availability of ammonia is substantially reduced in the R142I variant (as compared to WT ASNS) when Gln is used as the nitrogen source even though the kinetic parameters are unchanged in the ammonia-dependent activities of the two enzymes.

“This gating mechanism contrasts sharply with structural observations on other Class II amidotransferases for which substrate binding initiates large conformational rearrangements that lead to tunnel formation and activation of glutaminase activity. [...] MD simulations have confirmed that PRPP binding stabilizes the conformation of a key active site loop, which is disordered in the free enzyme.” This is what RMSFs plots seem to indicate in the case of ASNS reported here, so we disagree with this statement.

The very large rearrangements that are shown in Fig. S34 demonstrate that an ammonia tunnel is not present in the apo form of either GPA or GFAT. This clearly is different from our finding that the tunnel is present in the unliganded form of human ASNS, although it fluctuates between conformations. We therefore respectfully disagree with the reviewer’s assessment.

In Figure 6c, it is stated that a slate color is used for $t=0$ and $t=200$ ns. This should be corrected.

The figure legend has been corrected.

In all figures where a selected distance is displayed along the MD simulation time, the same color (or at least similar color) should be used for each replica (i.e., replica 1 always displayed using green color, replica 2 red). In this way one can easily capture if a given conformational change observed for a given distance is correlated to another one in the same replica.

The figures have been re-colored as suggested by the reviewer.

Reorder all figures in SI as they are cited in the main text.

We have re-ordered citations of the Supplementary Figures and Tables in the main text.

Indicate the exact atoms that are used to compute the distances in Figure S20 and S28, it is stated that the distance between Arg-142 and Glu-414 is displayed for the R142I mutant, so this should be corrected.

Typographical errors have been fixed in all of the legends for the Supplementary Figures, with several legends being re-written. We apologize for these oversights in the last submission.

Reviewer 5:

Q1: It is really surprising that a paper focused on the analysis of conformational dynamics by means of cryo-EM and MD contains an introduction with only a few citations to computational papers covering the analysis of the conformational landscapes.

A1: Almost all of these papers focus on allosteric networks, which generally impact substrate binding (K_m) rather than catalysis (k_{cat}). We agree, however, that these papers should be included and have revised the introductory text as requested by the reviewer. As a result, we have also added new citations (8, 13, 14, and 31-35).

Please note that one of the included references focuses on ImGPS enzyme, in which allostery mostly affects k_{cat} rather than K_m .

This comment seems unrelated to any statement in the main text of the manuscript and the SI, and therefore probably refers to the rebuttal statement above (A1). We have, as per the reviewer's request, cited relevant papers on this topic in the revised paper.

Q2: The authors analyze the effect of Arg-142 conformation as defined as open/closed (or in some cases partially open), on both the X-ray data and some snapshots along the MD simulations performed. How is this open/closed defined? Based on a distance, angle, dihedral of the Arg-142 sidechain? How do they know that Arg-142 conformation is directly related to tunnel formation? In the included Figures (4, 6) Arg-142 is not displayed, so such correlation cannot be extracted. It is therefore not clear how the authors know that Arg "open/closed" regulates tunnel formation.

A2: The definition of open and closed is based on whether the side chain blocks access to the tunnel (closed) or not (open), as seen in the EM structure. The key idea here is that Arg-142 gates the access of ammonia to the tunnel, whether the latter is in a continuous or discontinuous form. We have attempted to make this point clearer by using more precise language in the revised manuscript. We now also indicate the location of either the Arg-142 or Ile-142 side chain using red dots in Figures 4 and 6. A quantitative description of the two conformations is also now presented in Supplementary Figure S15, which forms the basis of our choice for the two collective variables used in the new metadynamics simulations (Supplementary Figure S21).

The mean and standard deviations of the set of collective variables identified and represented in Figure S15 (and subsequent ones, especially in Figure S21) should be included. What remains to be shown is that tunnel formation is regulated by Arg-142, which are the residues that form the bottleneck area of the computed CAVER tunnels? In those snapshots shown in Figures 4 and 6, the values of distances of Arg-142 and surrounding residues should be included, especially those distances used for the metadynamics simulations. I am also curious about the selection of the two CVs for performing the metadynamics simulations, both CVs are based on the distance between Arg-142 and either Glu-414 or Asp-405, so they are dependent on each other. More details should be provided on the convergency of metadynamics simulations.

We have included the requested information concerning average distances between Arg-142 and surrounding residues for the 200 ns replicates of the trajectories involving the WT enzyme (Supplementary Tables S2 and S4). In addition, we have listed specific distances for all snapshots in Figure 5 (originally Figure 4) in the supporting information (Supplementary Tables S3 and S5). Snapshots in Figure 6 are for the Ile-142 variant for which the side chain does not adopt different conformations.

While putting this information together, we finally understood the reviewers' concerns about our interpretations of the MD simulations and made substantial textual revisions to clarify matters. In particular, we make clear that Arg-142 can act to regulate ammonia access to the tunnel, and that formation of a continuous tunnel depends on a more complex set of motions involving multiple residues. We emphasize this point by stating in the text *"the gating function of Arg-142 is separate from the molecular events that form the continuous form of the tunnel; ammonia therefore travels to the synthetase site only if the tunnel is continuous and the Arg-142 side chain adopts the open conformation"*.

We thank the reviewers for pointing out these problems in our last submission and hope that they will agree with our modified conclusions.

Additional details requested for the metadynamics calculations (performed only to get a sense of the free energy barriers to moving the Arg-142 side chain) are included in a new figure (Supplementary Figures S35 and S36). As the two distances chosen for the CVs are completely independent of each other in the simulations, being chosen merely to distinguish between different conformations of the Arg-142 side chain, we do not believe that they are dependent on each other.

Q5: "Using the Robetta server, we constructed a model of full-length human apo-ASNS". It is not clear to us why the more recent and accurate AF2 multimer tool was not used to reconstruct the full model. How reliable is this Robetta structure?

A5: AF2 was not available when we performed the original simulations, and we sought to be consistent in subsequent calculations used to prepare the revised paper. We merely note that the Robetta server is widely thought to generate reliable models.

We understand that when the original simulations were done AF2 was not available, still it will be good to generate now the model with AF2 and compare it with the one obtained with Robetta. This will be useful to validate and justify the used model. This would also be useful to justify this sentence from the main text: "Equally, the large fluctuations of the C-terminal tail in the replicated MD trajectories most likely are a consequence of these residues being mispositioned in the initial model (t = 0 ns)"

We have included a comparison of our Robetta structure with the structure predicted by the AF3 in Supplementary Figure S17.

Q7: The authors performed MD simulations on both the monomeric and dimeric forms of the enzymes. However, what is the oligomerization state of the enzyme in solution? This could be determined experimentally, which will help then deciding which model is best to analyze and include in the main text.

A7: It is common practice to use a single monomer of a multimeric system in order to ensure that the simulations are completed in a reasonable amount of time and to facilitate analysis of the results. The oligomerization state of the enzyme in solution is irrelevant in our opinion because we are comparing the MD simulations to data from the EM-derived dimer structure.

Being experts in MD simulations, we can state that what it is a common practice is to run the MD simulations using the oligomerization state of the enzyme that adopts in solution. In fact, the MD simulations are performed in explicit water solvent, so to rationalize and get insights about functional states of the enzyme MD simulations should be performed using the set of conditions

that most closely matches the experimental conditions. This also applies to oligomerization states, although we agree that the analysis effort and computational cost is higher. One of the strong points of MD simulations is that it is complementary and provides additional insights to what is observed in X-ray and EM-derived structures.

There is every reason to believe that the biologically active form of human ASNS is a dimer as seen in the cryo-EM structure (as well as crystal structure) and we believe that we have provided evidence to support our choice of using the monomer in our MD simulations. One of us (Richards) has been studying this enzyme experimentally for over 30 years and has never seen any evidence to refute the idea that human ASNS forms anything other than monomers and dimers.

Q8: In Figure S18, a comparison of backbone RMSF values computed from 200 ns MD trajectories of the full-length apo-ASNS monomer and the apo-ASNS dimer is displayed. Although the authors mention that no significant differences were observed between monomer and dimer, the RMSF displayed in SI does show significant differences: the dimer appears to be much more flexible than the monomer, especially in the 480-540 region for monomer B.

A8: The differences between the dimer and the monomer are not significant (up to 1 Å is not really a difference). The most noticeable difference is indeed in the 480-540 region which is known to be highly flexible and anyway has no impact on the behavior of Arg-142.

In Figure S19 (now), differences of up to 2 Å can be observed in the 40-60 region (monomer A), and differences higher than 3 Å are observed in the 490-510 region (monomer B). The point that the flexibility of the 480-540 region has no impact on the Arg-142 behavior is not demonstrated.

There is no reason to believe that this region will have any impact on Arg-142 side chain behavior, which seems to be controlled by local interactions involving the salt bridge networks already shown and discussed in the manuscript. In any event, without actual X-ray or cryo-EM structures that show the precise location of this region, addressing the reviewers' assertion is unfortunately not feasible. Attempts to resolve this issue by comparing the behavior of models created using structure prediction tools clearly lie outside the scope of this paper, which, we remind the reviewer, is to demonstrate the utility of 3DVA to identify the conformational properties of functionally interesting residues in multi-domain enzymes.

Q9: In Figure 5, large differences between the cryo-EM structures and MD data are also observed in the region around 475-500, and 200-230. Why is that? Where are the regions located and how important are for the catalysis? The same is observed when comparing apo and ternary complex MD simulations, thus indicating that these regions play a key role for binding and catalysis. The authors mention that the large deviations correspond to regions not solved in the X-ray structure, so again we wonder about the reliability of the generated model.

A9: We thank the reviewer for pointing out this problem in Figure 5. These two regions are known to be flexible, and indeed, the structural models derived from either X-ray crystallography or cryoEM show low resolution at these sections of the enzyme. As a result, the fluctuation differences between the 3DVA and MD-derived arise from the lower quality of the EM-derived density for residues adjacent to residues that are missing in the EM structure. We have added new text and modified the figure legend to clarify what is going on.

We understand that these regions are not solved or have a low resolution, but we are still wondering about the role of these regions in binding and catalysis.

No experimental evidence is available that permits a resolution of this query. Obtaining such evidence and evaluating it with additional MD and metadynamics simulations is a separate study that lies outside the scope of the present manuscript.

Q13: The comparison of the tunnels between WT and variants is also not conclusive. In Figure 6b, the tunnel for the R142I variant is displayed, which shows that at the 20 and 140 ns snapshots there is actually no tunnel. They claim that the conformation of Ile is not regulating tunnel formation, so clearly the formation of the tunnel is affected by additional residues not considered in the manuscript. The analysis of the bottleneck of the tunnel and which residues are located close to this narrowest region might be helpful to provide additional insights.

A13: We have modified the text to clarify our interpretation. The key point that we were trying to convey was that the presence of the intermediate results in large structural changes in the tunnel even though the Ile-142 side chain adopts a conformation that would otherwise allow ammonia to access the tunnel. Given the reservations of the reviewer about the validity of these computational models, however, we consider that detailing the roles of multiple residues in determining the structural stability of the ammonia tunnel lies outside the scope of the paper. Moreover, omitting this discussion does not impact our key observation that EM can be used to observe the Arg-142 side chain in multiple conformations.

We disagree. The paper puts a lot of emphasis on the role of Arg-142 sidechain conformation in tunnel formation, and in fact, the authors have already performed CAVER calculations, so that the information on the residues defining the bottleneck of the computed tunnels is already included in the performed calculations. This information will provide very relevant information and will confirm their hypothesis based on a set of distances and visual inspection of the structures and MD data.

While we can readily appreciate the reviewer's assertion that detailing the roles of multiple residues by MD stimulations will provide additional insights into tunnel structure and dynamics, such an analysis lies outside the scope of the present study and is best addressed in a separate paper that will appear in a more specialized journal. Nonetheless, we have acknowledged this point by including additional text in the Discussion section of this revision of the manuscript. In the meantime, interested readers will be able to access the PDB coordinate files used to make the snapshots as part of our obligations under open access publishing.

Reviewer 5:

In this new revised version of the manuscript, the authors have addressed our original concerns (but not all of them).

We sincerely thank the reviewer for acknowledging that the revisions made in the 3rd revision of the paper have addressed their scientific critiques. As we outline below, their remaining concerns seem to be very minor and, in our view, do not negatively affect the central scientific points we are reporting in this manuscript.

The comparison of the Robetta and AF3 model included in S17 shows large differences at the C-terminal region, but also at other loops (201-220, 465-475) contained in the middle and end of the sequence. Unfortunately, none of these regions are solved in the X-ray or cryo-EM structure, which indicates that these are highly dynamical regions (in the RMSF plots these regions display also some flexibility). However, the large deviation of both structures again questions the validity of the Robetta model, as AF2 has shown to predict highly accurate folded structures especially if similar structures were included in the training set.

As the reviewers rightly point out, the two predicted structures differ **precisely** in the regions of the enzyme that are not visible in experimental data. This observation aligns with our view that these regions are conformationally flexible, particularly the C-terminal ends where the loop is followed by an alpha-helical structure. Notably, the predicted alpha-helical structures from both Robetta and AF3 are almost identical; the key difference lies in the positioning of the alpha-helical structure. Unlike the loops (aa201-220, aa465-475), where the ends are structurally constrained, and show little variation between AF3 and Robetta, the C-terminal end is more flexible and can move freely, which explains the variation in the predictions. As we cannot directly validate either initial model due to a lack of experimental data, it is not a matter of choosing a "correct" model for our MD simulations. In any case, we emphasize that our MD simulations were designed to complement our observations regarding the functional importance of Arg-142, as established by 3DVA, mutagenesis, and kinetic assays, none of which have been questioned by the reviewers. Therefore, we believe the issue raised above is more a matter of opinion or preference rather than identifying a fundamental scientific problem with our study.

Even if we were to repeat the calculations using the AF3 model, the outcome would either align with the Robetta-based results or differ. In the latter case, we would find ourselves in a position where it is impossible to determine which result is correct based on the available experimental evidence. Currently, the validity of our Robetta-based calculations is evidenced by the fact that our simulations produce RMSF values which closely match those derived from 3DVA-derived (i.e., experimental) structures (Fig. 4). Our simulations are also consistent with the clear, experimentally observed impact of the aspartyl-AMP intermediate on the conformational energetics of the Arg-142 side chain, as shown by our metadynamics simulations (Fig. S19). Such consistencies would not be possible if the Robetta-based model were entirely inaccurate. In light of the award of the Nobel Prize in Chemistry to David Baker and the team behind DeepMind's AlphaFold, it seems that the Robetta server has been recognized to be at least as reliable as AlphaFold3.

In summary, our calculations are consistent with experimental data. Given this, we see no scientific benefit to repeating all of our MD simulations using the AF3 model for two main reasons: (i) there is no experimental evidence to suggest that the AF3 model is more accurate, and (ii) most importantly, the outcome of this additional work will not change the central conclusion of the paper - **that 3D variability analysis of a cryo-EM map can provide insights at the level of**

single side chains and generate hypotheses regarding the functional importance of individual residues in the enzyme.

If there is every reason to believe that the biologically active form is a dimer, there is no need to run the MD simulations with the monomeric form of the enzyme.

This concern appears to reflect the reviewer's perspective on MD simulations rather than a substantive issue that affects the key conclusions of our study, which are based on uncontested experimental observations. As we highlighted in previous responses to reviewers, the MD simulations reported in this manuscript were conducted to (i) validate the conformational changes of the Arg-142 side chain identified through 3D variability analysis, and (ii) provide qualitative insights into the intermolecular interactions that change as the Arg-142 side chain adopts different conformations. The impact of these movements on tunnel continuity was unexpected and led us to formulate hypotheses, which were subsequently tested using the R142I variant.

Performing the MD simulations with dimer rather than the monomer does not alter our conclusions regarding replacement of arginine with isoleucine, and the observed changes in the kinetic properties of the R142I variant are fully consistent with our simulations. As importantly, we show that the monomeric components of the ASNS dimer exhibit motions that are essentially similar to those of an isolated monomer in solution (Fig. S37). We therefore believe that our qualitative insights into the functional role of Arg-142 drawn on the basis of simulations on the ASNS monomer will be essentially unchanged compared to those obtained from more computationally demanding simulations of the ASNS dimer. The reviewer is free to disagree with our interpretations but must recognize that this difference of opinion does not affect the significance and validity of using 3DVA to generate hypotheses about residue function and conformational preferences from cryo-EM maps.

As the authors know, conformational changes/mutations/effector binding/etc far away from the active site can have a large influence on enzyme function. It might very well be that this region has no effect on Arg conformation, but considering the allosteric nature displayed by many enzymes, there might be some reasons to believe otherwise. In any case, this region is very close to the 465-475 region that Robetta/AF3 predicted different conformations, again questioning the validity of the used model.

We are unable to confirm or refute the reviewer's perspective regarding allostery in human ASNS, because no experimental studies have been reported that either support or dismiss the existence of allosteric networks in this enzyme. Again, although we appreciate the reviewer's viewpoint, we would like to reiterate that this criticism of our work is a matter of opinion and does not therefore affect the validity of our experimental findings or the central conclusions of the study. As noted earlier, repeating the entire set of calculations using the alternative AF3-derived model is likely be an unnecessary use of our computational resources.

As mentioned in our previous report, the authors have already computed the tunnels with CAVER, so the information of the residues defining the bottleneck of the computed tunnels is already included in their already performed calculations. Therefore, no additional calculations need to be performed, but rather check the bottleneck information in the output.

As discussed in our rebuttal submitted with the 3rd revision of the paper, we believe that a detailed analysis of the tunnel residues is beyond the scope of the current manuscript. Furthermore, if the reviewer feels that the MD trajectories are compromised due to the use of an incorrect initial model, it is difficult to see how including information about the "bottleneck" residues would be

meaningful. On the contrary, providing this information would seem to suggest that our simulations are valid, thereby undermining any concerns about the choice of model.

Given that the trajectory data will be made available to all readers (we will cover the open access charges and ensure compliance with *Nature Communications* policies), anyone interested will have the opportunity to obtain this information themselves. Additionally, we do not believe that including the requested information would alter the validity of our conclusions regarding 3DVA, which is the primary focus of this manuscript.

Reviewer #5 attachment:

The authors have substantially revised and modified the manuscript following the referees' concerns. However, many of the raised concerns have not been properly addressed, and therefore we cannot recommend publication of the paper unless the following comments are considered.

Q1: It is really surprising that a paper focused on the analysis of conformational dynamics by means of cryo-EM and MD contains an introduction with only a few citations to computational papers covering the analysis of the conformational landscapes.

A1: Almost all of these papers focus on allosteric networks, which generally impact substrate binding (K_m) rather than catalysis (k_{cat}). We agree, however, that these papers should be included and have revised the introductory text as requested by the reviewer. As a result, we have also added new citations (8, 13, 14, and 31-35).

Please note that one of the included references focuses on ImGPS enzyme, in which allostery mostly affects k_{cat} rather than K_m .

Q2: The authors analyze the effect of Arg-142 conformation as defined as open/closed (or in some cases partially open), on both the X-ray data and some snapshots along the MD simulations performed. How is this open/closed defined? Based on a distance, angle, dihedral of the Arg-142 sidechain? How do they know that Arg-142 conformation is directly related to tunnel formation? In the included Figures (4, 6) Arg-142 is not displayed, so such correlation cannot be extracted. It is therefore not clear how the authors know that Arg "open/closed" regulates tunnel formation.

A2: The definition of open and closed is based on whether the side chain blocks access to the tunnel (closed) or not (open), as seen in the EM structure. The key idea here is that Arg-142 gates the access of ammonia to the tunnel, whether the latter is in a continuous or discontinuous form. We have attempted to make this point clearer by using more precise language in the revised manuscript. We now also indicate the location of either the Arg-142 or Ile-142 side chain using red dots in Figures 4 and 6. A quantitative description of the two conformations is also now presented in Supplementary Figure S15, which forms the basis of our choice for the two collective variables used in the new metadynamics simulations (Supplementary Figure S21).

The mean and standard deviations of the set of collective variables identified and represented in Figure S15 (and subsequent ones, especially in Figure S21) should be included. What remains to be shown is that tunnel formation is regulated by Arg-142, which are the residues that form the bottleneck area of the computed CAVER tunnels? In those snapshots shown in Figures 4 and 6, the values of distances of Arg-142 and surrounding residues should be included, especially those distances used for the metadynamics simulations. I am also curious about the selection of the two CVs for performing the metadynamics simulations, both CVs are based on the distance between Arg-142 and either Glu-414 or Asp-405, so they are dependent on each other. More details should be provided on the convergency of metadynamics simulations.

Q5: “Using the Robetta server, we constructed a model of full-length human apo-ASNS”. It is not clear to us why the more recent and accurate AF2 multimer tool was not used to reconstruct the full model. How reliable is this Robetta structure?

A5: AF2 was not available when we performed the original simulations and we sought to be consistent in subsequent calculations used to prepare the revised paper. We merely note that the Robetta server is widely thought to generate reliable models.

We understand that when the original simulations were done AF2 was not available, still it will be good to generate now the model with AF2 and compare it with the one obtained with Robetta. This will be useful to validate and justify the used model. This would also be useful to justify this sentence from the main text: “Equally, the large fluctuations of the C-terminal tail in the replicated MD trajectories most likely are a consequence of these residues being mispositioned in the initial model (t = 0 ns)”

Q7: The authors performed MD simulations on both the monomeric and dimeric forms of the enzymes. However, what is the oligomerization state of the enzyme in solution? This could be determined experimentally, which will help then deciding which model is best to analyze and include in the main text.

A7: It is common practice to use a single monomer of a multimeric system in order to ensure that the simulations are completed in a reasonable amount of time and to facilitate analysis of the results. The oligomerization state of the enzyme in solution is irrelevant in our opinion because we are comparing the MD simulations to data from the EM-derived dimer structure.

Being experts in MD simulations, we can state that what it is a common practice is to run the MD simulations using the oligomerization state of the enzyme that adopts in solution. In fact, the MD simulations are performed in explicit water solvent, so to rationalize and get insights about functional states of the enzyme MD simulations should be performed using the set of conditions that most closely matches the experimental conditions. This also applies to oligomerization states, although we agree that the analysis effort and computational cost is higher. One of the strong points of MD simulations is that it is complementary and provides additional insights to what is observed in X-ray and EM-derived structures.

Q8: In Figure S18, a comparison of backbone RMSF values computed from 200 ns MD trajectories of the full-length apo-ASNS monomer and the apo-ASNS dimer is displayed. Although the authors mention that no significant differences were observed between monomer and dimer, the RMSF displayed in SI does show significant differences: the dimer appears to be much more flexible than the monomer, especially in the 480-540 region for monomer B.

A8: The differences between the dimer and the monomer are not significant (up to 1 Å is not really a difference). The most noticeable difference is indeed in the 480-540 region which is known to be highly flexible and anyway has no impact on the behavior of Arg-142.

In Figure S19 (now), differences of up to 2 Å can be observed in the 40-60 region (monomer A), and differences higher than 3 Å are observed in the 490-510 region (monomer B). The point that the flexibility of the 480-540 region has no impact on the Arg-142 behaviour is not demonstrated.

Q9: In Figure 5, large differences between the cryo-EM structures and MD data are also observed in the region around 475-500, and 200-230. Why is that? Where are the regions located and how important are for the catalysis? The same is observed when comparing apo and ternary complex MD simulations, thus indicating that these regions play a key role for binding and catalysis. The authors mention that the large deviations correspond to regions not solved in the X-ray structure, so again we wonder about the reliability of the generated model.

A9: We thank the reviewer for pointing out this problem in Figure 5. These two regions are known to be flexible, and indeed, the structural models derived from either X-ray crystallography or cryoEM show low resolution at these sections of the enzyme. As a result, the fluctuation differences between the 3DVA and MD-derived arise from the lower quality of the EM-derived density for residues adjacent to residues that are missing in the EM structure. We have added new text and modified the figure legend to clarify what is going on.

We understand that these regions are not solved or have a low resolution, but we are still wondering about the role of these regions in binding and catalysis.

Q13: The comparison of the tunnels between WT and variants is also not conclusive. In Figure 6b, the tunnel for the R142I variant is displayed, which shows that at the 20 and 140 ns snapshots there is actually no tunnel. They claim that the conformation of Ile is not regulating tunnel formation, so clearly the formation of the tunnel is affected by additional residues not considered in the manuscript. The analysis of the bottleneck of the tunnel and which residues are located close to this narrowest region might be helpful to provide additional insights.

A13: We have modified the text to clarify our interpretation. The key point that we were trying to convey was that the presence of the intermediate results in large structural changes in the tunnel even though the Ile-142 side chain adopts a conformation that would otherwise allow ammonia to access the tunnel. Given the reservations of the reviewer about the validity of these computational models, however, we consider that detailing the roles of multiple residues in determining the structural stability of the ammonia tunnel lies outside the scope of the paper. Moreover, omitting this discussion does not impact our key observation that EM can be used to observe the Arg-142 side chain in multiple conformations.

We disagree. The paper puts a lot of emphasis on the role of Arg-142 sidechain conformation in tunnel formation, and in fact, the authors have already performed CAVER calculations, so that the information on the residues defining the bottleneck of the computed tunnels is already included in the performed calculations. This information will provide very relevant information and will confirm their hypothesis based on a set of distances and visual inspection of the structures and MD data.